# Human caspase-4 detects tetra-acylated LPS and cytosolic *Francisella* and functions differently from murine caspase-11

Brice Lagrange[1], Sacha Benaoudia[1], Pierre Wallet[1], Flora Magnotti[1], Angelina Provost[1], Fanny Michal[1], Amandine Martin[1], Flaviana Di Lorenzo[2], Bénédicte F. Py[1], Antonio Molinaro[2] & Thomas Henry [1]

Caspase-4/5 in humans and caspase-11 in mice bind hexa-acylated lipid A, the lipid moeity of lipopolysaccharide (LPS), to induce the activation of non-canonical inflammasome. Pathogens such as *Francisella novicida* express an under-acylated lipid A and escape caspase-11 recognition in mice. Here, we show that caspase-4 drives inflammasome responses to *F. novicida* infection in human macrophages. Caspase-4 triggers *F. novicida*-mediated, gasdermin D-dependent pyroptosis and activates the NLRP3 inflammasome. Inflammasome activation could be recapitulated by transfection of under-acylated LPS from different bacterial species or synthetic tetra-acylated lipid A into cytosol of human macrophage. Our results indicate functional differences between human caspase-4 and murine caspase-11. We further establish that human Guanylate-binding proteins promote inflammasome responses to under-acylated LPS. Altogether, our data demonstrate a broader reactivity of caspase-4 to under-acylated LPS than caspase-11, which may have important clinical implications for management of sepsis.

[1] CIRI, Centre International de Recherche en Infectiologie, Inserm, U1111, Université Claude Bernard Lyon 1, CNRS, UMR5308, École Normale Supérieure de Lyon, Univ Lyon, F-69007 Lyon, France. [2] Department of Chemical Sciences, University of Napoli Federico II, Complesso Universitario Monte Santangelo, Via Cintia 4, I-80126 Napoli, Italy. Correspondence and requests for materials should be addressed to T.H. (email: thomas.henry@inserm.fr)

The inflammasomes form one of the first lines of defense of the innate immune system to fight invading pathogens. The canonical inflammasomes are multimolecular platforms for caspase-1 activation[1]. Inflammasome activation relies on the sensing of pathogen-associated molecular patterns (PAMP) or damage-associated molecular patterns (DAMP) by specific pattern recognition receptors (PRR). Active caspase-1 cleaves the pro-inflammatory cytokine precursors, pro-interleukin-1β (IL-1β), and pro-interleukin-18 (IL-18)[2], for their eventual release from the cells. Caspase-1 also triggers pyroptosis through the cleavage of gasdermin D[3, 4]. Although activation of caspase-1 defines the canonical inflammasomes, the non-canonical inflammasome involves caspase-1-independent activation of the murine caspase-11, or of its human orthologs caspase-4/5[5]. Caspase-4 and caspase-5 cleave gasdermin D and thus trigger pyroptosis. The gasdermin D-mediated membrane damage that is associated with non-canonical inflammasome activation subsequently induces NLRP3 inflammasome activation[5–8].

Different inflammasome complexes engage different PRRs depending on the nature of the detected PAMPs or DAMPs[9]. Several inflammasomes can be implicated in the detection of a specific pathogen[10–12]. Using bone marrow-derived macrophages (BMDM) and gene deletion, inflammasome-mediated defenses against many pathogenic bacteria have been documented in mice[9]. However, much less information is available regarding human inflammasome activation. Furthermore, there is evidence that the inflammasome responses can differ between mice and humans[13–15], as demonstrated using *Francisella novicida*.

*F. novicida* is a close relative of *F. tularensis*, the agent of tularemia[16, 17]. *F. novicida* virulence is linked to its ability to escape from the macrophage phagosome and replicate within the host cytosol. *F. novicida* has emerged as a tool to study cytosolic innate immune responses and, in particular, the inflammasome[17]. In murine wild-type (WT) macrophages infected with *F. novicida*, IL-1β release relies almost exclusively on the sensing of bacterial DNA by Aim2[18, 19]. In agreement with the exclusive sensing by Aim2, *F. novicida* escapes caspase-11 recognition due its tetra-acylated LPS[20, 21]. In contrast to what is observed in murine macrophages, both NLRP3 and AIM2 have been described to mediate IL-1β processing in phorbol 12-myristate 13-acetate (PMA)-differentiated THP-1 cells, a model for primary human macrophages[16]. In addition, Pyrin, an inflammasome sensor detecting RhoA GTPases inhibition[22], has been implicated in the immune responses to *F. novicida* infection in human monocytes and macrophages[23]. The function of the various inflammasome complexes in human macrophages is thus still unclear.

*F. novicida*-mediated Aim2 activation in murine macrophages depends on type I IFN signaling and several IFN-inducible proteins, including the Guanylate-binding proteins 2 (Gbp2) and Gbp5, as well as IrgB10[18, 19, 24–27]. IrgB10 and Gbps are both required to lyse *F. novicida*, release DNA into the host cytosol and activate the Aim2 inflammasome[24, 25, 27]. Interestingly, human GBPs differ from murine Gbps due to multiple gene amplifications in mice[28]. Furthermore, the Irg resistance system (including IrgB10) has been lost in humans[29]. The relevance of human GBPs and their function in *F. novicida* immune defense thus remain to be characterized.

To study the human inflammasome complexes and co-factors implicated in sensing cytosolic *F. novicida*, we investigate the contribution of inflammasome sensors, inflammatory caspases and GBPs in primary human monocyte-derived macrophages (hMDM). We demonstrate an essential function of the non-canonical caspase-4 inflammasome in human macrophages, which implicates an intrinsic difference between the functionality of caspase-4 and caspase-11. Importantly, this difference is also evident upon direct delivery of synthetic tetra-acylated lipid A or under-acylated LPS from *Francisella* and other Gram-negative bacteria to primary human macrophages. Although the observed response to under-acylated LPS/lipid A is weaker than the response to hexa-acylated LPS/lipid A, these results indicate that the human non-canonical inflammasome detects a larger diversity of LPS molecules than its murine counterpart does. Furthermore, we show that GBP2 contributes to activation of the non-canonical inflammasome in human macrophages, demonstrating that the function of GBPs in innate immune defenses against intracellular bacteria is conserved from mice to men.

## Results

**The FPI is required for inflammasome activation in hMDMs**. *F. novicida* escape from the phagosome is required to trigger inflammasome activation in BMDMs[17] and human monocytes[30]. However, inflammasome activation is poorly characterized in hMDMs. To assess the impact of *Francisella* phagosomal escape on inflammasome activation, we infected hMDMs with WT *F. novicida* or a mutant lacking the *Francisella* pathogenicity island (FPI). The FPI encodes an atypical type 6 secretion system (T6SS) essential for escape from the phagosome[31]. As previously observed in murine BMDMs[32], *F. novicida* triggered IL-1β and IL-18 release in a FPI- and time-dependent manner (Fig. 1a; Supplementary Fig. 1a). Similarly, hMDMs infected with WT *F. novicida* died rapidly while infection with the ΔFPI mutant did not substantially affect macrophage survival even at high multiplicity of infection (Fig. 1b; Supplementary Fig. 1b). Cell death, IL-1β and IL-18 release were associated with cleaved caspase-1 p20 in the supernatant of macrophages (Fig. 1c) indicating that cytosolic sensing of *F. novicida* triggers caspase-1 activation and processing in human macrophages. Interestingly, although IL-1α secretion is described to be largely caspase-1-independent[33, 34], IL-1α secretion followed the same pattern as IL-18 and IL-1β and was strictly dependent on the presence of the *F. novicida* into the host cytosol (Fig. 1a; Supplementary Fig. 1a). Of note, each strain induced a similar tumor necrosis factor (TNF) response (Fig. 1a; Supplementary Fig. 1a) demonstrating the specificity of inflammasome activation and IL-1α release in regards to *F. novicida* cytosolic localization.

**NLRP3 controls cytokine release in infected hMDMs**. *F. novicida*-mediated IL-1β, IL-α release and caspase-1-mediated cell death are fully dependent on Aim2 in murine BMDMs[18, 19] (Supplementary Fig. 2). In contrast, three different sensors (Pyrin, AIM2, and NLRP3) have been implicated in the detection of cytosolic *F. novicida* in various human phagocytes and cell lines[16, 23]. The relevance of these sensors in mediating inflammasome activation in hMDMs is unclear[17]. Although the *MEFV* transcript (encoding Pyrin) was greatly up-regulated during *F. novicida* infection (Supplementary Fig. 3a), we did not observe any impact of *MEFV* knockdown (Supplementary Fig. 3b) on *F. novicida*-mediated hMDMs responses (Supplementary Fig. 3c, d). As expected, *MEFV* knockdown highly reduced IL-1β secretion in response to *Clostridium difficile* toxin B (TcdB) (Supplementary Fig. 3e). The lack of Pyrin implication in sensing *F. novicida* in macrophages was confirmed in the human monocyte/macrophage cell line U937. Indeed, U937 cells overexpressing Pyrin demonstrated increased IL-1β secretion upon TcdB treatment but similar response as Pyrin-deficient U937 cells upon *F. novicida* infection or Nigerin treatment (Supplementary Fig. 3f–i).

In contrast, *NLRP3* knockdown (Supplementary Fig. 3b) had a major impact on IL-1β, IL-18, and IL-1α (Fig. 2a) indicating its central function in promoting cytokine release in *F. novicida*-infected hMDMs. Similarly, *NLRP3* knockdown strongly reduced caspase-1 processing and the release of caspase-1 p20 and mature

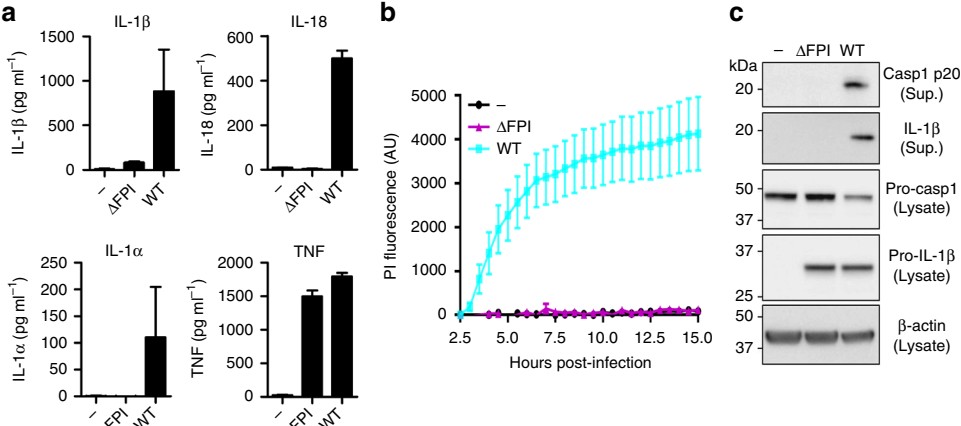

**Fig. 1** *F. novicida* FPI-encoded T6SS is required to trigger inflammasome activation in hMDMs. **a–c** hMDMs were infected at a multiplicity of infection of 10 with wild-type (WT) *F. novicida* or a mutant deleted of the *Francisella* pathogenicity island (FPI) encoding the type VI secretion system (T6SS). **a** Interleukin (IL)-1β, IL-18, IL-1α, and tumor necrosis factor (TNF) levels were quantified by ELISA at 9 h post-infection. **b** Cell death was assessed in real-time by measuring propidium iodide (PI) incorporation/fluorescence every 30 min (AU arbitrary units). **c** Caspase-1 and IL-1β processing and protein levels were evaluated in the supernatant (Sup.) and the lysate of hMDMs infected or not for 9 h. **a**, **b** Values represent the mean ± standard deviation from two to three independent experiments. **c** One experiment representative of two independent experiments is shown

IL-1β in the supernatant (Fig. 2b). Conversely, *AIM2* knockdown (Supplementary Fig. 3b) had no impact on IL-1 release (Fig. 2a). *AIM2* knockdown had a minor and non-significant (although observed with 7 out of 8 donors; $p > 0.05$, by *t*-test) effect on IL-18 levels (Fig. 2a). Although *NLRP3* knockdown led, on average, to a 73% decrease in IL-18 levels in the supernatant, *AIM2* knockdown led, on average, to a 24% reduction in IL-18 levels. These results confirmed a previous study performed by Harton and colleagues in THP-1 macrophages demonstrating striking differences in the hierarchy of the AIM2 and NLRP3 inflammasomes in human versus murine macrophages[16]. Furthermore, the dual dependency on NLRP3 and AIM2 to get a full IL-18 response in hMDMs mirrors, although with an inverted ratio, what has been observed in *F. novicida*-infected murine BMDMs[35]. Surprisingly, neither *NLRP3* nor *AIM2* knockdown had any effect on cell death levels as measured by lactate dehydrogenase (LDH) release (Fig. 2c) or propidium iodide incorporation (Supplementary Fig. 3c). To exclude any bias associated with siRNA-mediated knockdown, we used the Crispr/Cas9 technology to invalidate *AIM2* or *NLRP3* in U937 cells (Supplementary Fig. 4). The obtained cell lines further confirmed that NLRP3 but not AIM2 strongly contributed to caspase-1 maturation and release, IL-1β processing and release upon *F. novicida* infection (Fig. 2d, e). Neither of the two inflammasome molecules contributed to *F. novicida*-mediated U937 cell death as assessed by monitoring LDH release (Fig. 2f) or propidium iodide influx (Fig. 2g). Thus, while NLRP3 controlled IL-1α, IL-1β, and IL-18 release (the latter with a possible contribution of AIM2), none of the canonical inflammasome sensors considered here were involved in programming *F. novicida*-infected hMDM death.

**Caspase-4 controls hMDMs responses to *F. novicida* infection.** As hMDMs death was independent of NLRP3, AIM2 and Pyrin, we assessed the relative contribution of the canonical and non-canonical inflammasomes in hMDMs responses. Caspase-1 transcript was constitutively expressed. We observed an induction of the expression of two other inflammatory caspases, caspase-4 and 5 upon infection (Fig. 3a; Supplementary Fig. 5a), which was confirmed by immunoblot analysis (Fig. 3b;

Supplementary Fig. 5b). Interestingly, knockdown of caspase-4 (Supplementary Fig. 3b) strongly decreased IL-1β, IL-18, and IL-1α levels in *F. novicida*-infected hMDMs, whereas it had no impact on TNF levels (Fig. 3c). IL-1β and IL-18 release were reduced upon caspase-1 knockdown while we did not observe any impact on IL-1α release (Fig. 3c). The function of the inflammatory caspases in IL-1β release was strengthened using z-YVAD-FMK and z-LEVD-FMK inhibitors (Supplementary Fig. 6) although we could not ascertain whether each inhibitor specifically inhibited caspase-1 and/or caspase-4[36]. These results indicate that caspase-4, contrary to its murine ortholog caspase-11 (Supplementary Fig. 2), is the main effector of the cytosolic innate immune responses to *F. novicida*. Importantly, caspase-4 also controlled *F. novicida*-mediated hMDMs death (Fig. 3d; Supplementary Fig. 7). We did not observe any function for caspase-1 in mediating death further demonstrating the discrepancy with the results observed in murine BMDMs (Supplementary Fig. 2b). Caspase-4/gasdermin D-mediated pyroptosis and its associated $K^+$ efflux trigger NLRP3 inflammasome activation[7, 8]. In agreement with caspase-4 acting upstream of the canonical NLRP3 inflammasome, knockdown of caspase-4 decreased caspase-1 activation as assessed by immunoblot analysis of caspase-1 p20 subunit in the supernatant (Fig. 3e—88% reduction). As expected knockdown of caspase-4 did not affect pro-caspase-1 level in the cell lysate, whereas knockdown of caspase-1 led to a 62% reduction of pro-capase-1 protein level. The efficacy of caspase-1 siRNA to decrease the level of casp1 p20 in the supernatant was slightly lower (58% reduction) possibly due to the number of active ASC complex being more limiting for caspase-1 maturation/release than pro-caspase-1 level. To further confirm the function of caspase-4, we invalidated *CASP1*, *CASP4* and *GSDMD* genes in the U937 cell line using CRISPR/Cas9 technology. As previously observed using siRNA-mediated knockdown in hMDMs, CRISPR/Cas9-mediated gene ablation demonstrated in PMA-differentiated U937 macrophages, a key function of caspase-4 and its downstream target gasdermin D in promoting *F. novicida*-mediated caspase-1 maturation and release, IL-1β processing and release and cell death (Fig. 3f–h). We did not observe clear processing of the endogenous pro-caspase-4 possibly due to the relatively low level of the protein. To circumvent this problem, we over-expressed 3×Flag-Caspase-4 in

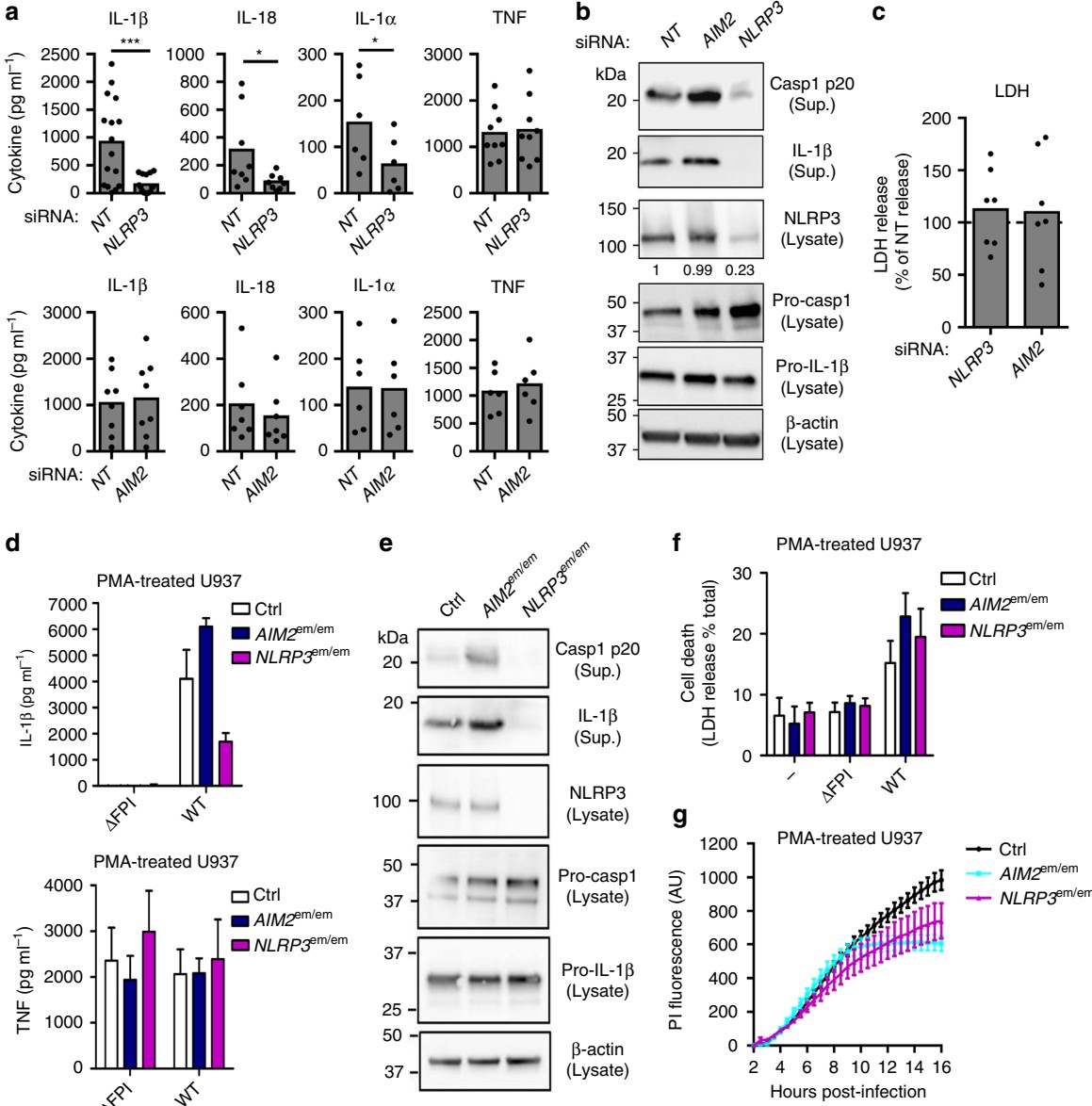

**Fig. 2** NLRP3 is required for *F. novicida*-mediated inflammasome activation in the cytosol of hMDMs. **a–c** hMDMs were transfected with non-targeting (NT) siRNA or with the indicated siRNA and infected with *F. novicida* at a multiplicity of infection (MOI) of 10 for 6 h (IL-1β, TNF and lactate dehydrogenase (LDH)) or 8 h (IL-18 and IL-1α, Western blot). **a** IL-1β, IL-18, IL-1α, and TNF levels were quantified by ELISA. **b** NLRP3, Caspase-1, and IL-1β protein level and processing were evaluated in the supernatant (Sup.) and the lysate of infected hMDMs. **c** Cell death was assessed by measuring LDH release. **d–g** IFN-γ-treated, phorbol 12-myristate 13-acetate- (PMA) differentiated control (Ctrl) U937 and U937 cell lines generated using CRISPR/Cas9 endonuclease-mediated (em) invalidation were infected with *F. novicida* at a MOI of 100 and monitored for **d** IL-1β and TNF release at 6 h post-infection or cell death by measuring **f** LDH release at 6 h post-infection and **g** propidium iodide (PI) incorporation/fluorescence in real-time (AU arbitrary units). **e** NLRP3 protein levels, caspase-1 and IL-1β processing were evaluated as indicated in the lysate and/or the supernatant (Sup.) of U937 cells infected for 8 h. **a, c** Each point shows the mean value from three technical replicates for one healthy donor, the bar shows the mean value for all donors ($n \geq 6$). Two-tailed $p$ values with the following nomenclature (***$p < 0.001$ and *$p < 0.05$ by paired $t$-test) are shown. **b, e** One experiment representative of two independent experiments is shown. **c** For each healthy donor, the mean value from cells transfected with NT siRNA was used as a reference value set to 100%. **d, f** Mean values ± standard deviation (SD) from three independent experiments are shown. **g** Mean values ± SD from three technical replicates from one experiment representative of three independent experiments are shown

U937 cells (Supplementary Fig. 8a). Overexpression of 3×Flag-Caspase-4 increased *F. novicida*-mediated LDH release (Supplementary Fig. 8b) and allowed a clear detection of caspase-4 processed form in the supernatant (Supplementary Fig. 8a). Importantly, the processed form was detected only upon infection with WT *F. novicida* but not upon infection with the ΔFPI mutant strongly suggesting that caspase-4 processing is associated

with cytosolic detection of *F. novicida*. Altogether, our data demonstrate that in human macrophages, cytosolic *F. novicida* sensing primarily depends on caspase-4 and on the non-canonical inflammasome. As previously described in other experimental systems[5–8], caspase-4/gasdermin D-mediated cell death triggers NLRP3-dependent caspase-1 activation and the release of IL-1β and IL-18.

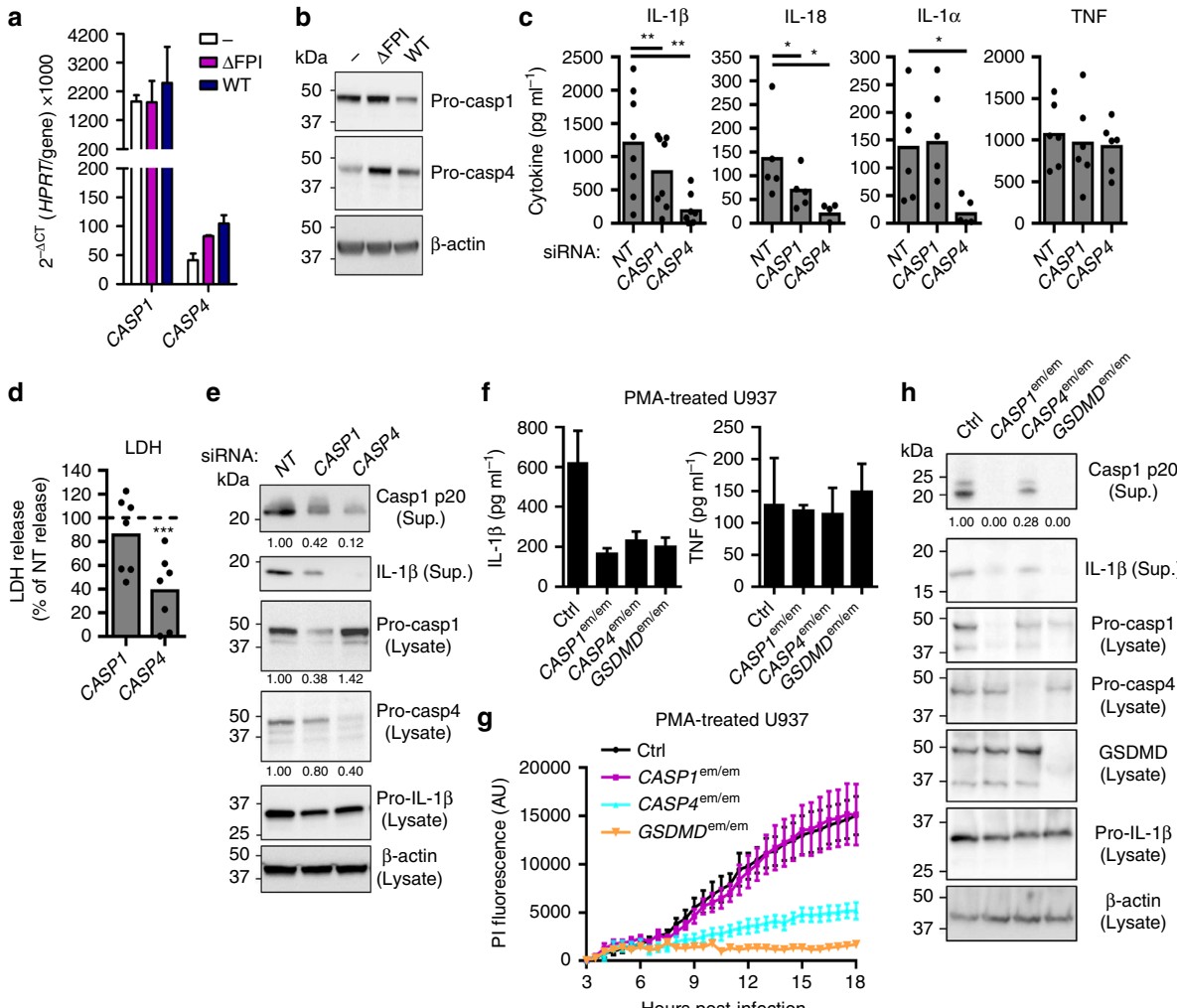

**Fig. 3** Caspase-4 is central to the inflammasome response of *F. novicida*-infected hMDMs. **a, b** Expression profile of caspases 1 and 4 by **a** quantitative RT-PCR and **b** Western blot analysis in hMDMs infected with wild-type (WT) *F. novicida* or the ΔFPI mutant at a multiplicity of infection (MOI) of 10 for 8 h. **c–e** hMDMs were transfected with non-targeting siRNA (NT) or with the indicated siRNA then infected with *F. novicida* at a multiplicity of infection (MOI) of 10 for 6 h (IL-1β, TNF, and lactate dehydrogenase (LDH)) or 8 h (IL-18 and IL-1α). **c** IL-1β, IL-18, IL-1α, and TNF levels in the supernatant were quantified by ELISA. **d** Cell death was assessed by measuring LDH release. **e** Caspase-4, caspase-1, and IL-1β protein levels and processing were evaluated from the supernatant (Sup.) or the lysate of hMDMs, 8 h post-infection. **f–h** IFN-γ-treated, phorbol 12-myristate 13-acetate- (PMA) differentiated control (Ctrl) U937 and U937 cell lines generated using CRISPR/Cas9 endonuclease-mediated (em) gene invalidation were infected with *F. novicida* at a MOI of 100 and monitored for **f** IL-1β and TNF release at 6 h post-infection or **g** cell death by propidium iodide (PI) incorporation/fluorescence (AU: arbitrary units). **h** *CASP1*, *CASP4*, and *GSDMD* gene invalidations, caspase-1 and IL-1β processing were evaluated in the lysate and the supernatant (Sup.) of infected cells for 8 h. **a, f** Mean values ± standard deviation (SD) from three independent experiments are shown. **b, e, h** One experiment representative of two independent experiments is shown. **c, d** Each point shows the mean value from three technical replicates for one healthy donor. The bar shows the mean value for all donors ($n \geq 5$). Two-tailed *p* values with the following nomenclature (***$p < 0.001$, **$p < 0.01$, and *$p < 0.05$ by paired *t*-test) are shown. **d** For each healthy donor, the mean value from cells transfected with NT siRNA was used as a reference value set to 100%. **g** Mean values ± SD from three technical replicates from one experiment representative of three independent experiments are shown

**F. tularensis LPS activates caspase-4 in hMDMs.** The strong caspase-4-dependency of the human macrophage responses to cytosolic *F. novicida* was unexpected. Indeed, Miao and collaborators[21] identified that caspase-11, the ortholog of caspase-4 in mice, was not activated by *F. novicida* or *F. tularensis* LPS transfection. Accordingly, we observed that transfection of *F. novicida* or *F. tularensis* Live Vaccine Strain (LVS) LPS in BMDMs did not elicit IL-1β secretion (Fig. 4a). As expected, *E. coli* LPS transfection, used as a control, activated caspase-11-dependent IL-1β secretion (Fig. 4a). We thus investigated whether hMDMs might be more sensitive than BMDMs to cytosolic *Francisella* LPS. Transfection of *F. tularensis* or *F. novicida* LPS

triggered caspase-1 maturation and release (Supplementary Fig. 9a) and IL-1β release in hMDMs (Fig. 4b). IL-1β levels were dependent on both caspase-4 and caspase-1 (Fig. 4c) demonstrating that caspase-4 senses cytosolic *F. novicida* LPS to trigger inflammasome activation. Of note, we observed a low level of IL-1β release in hMDMs incubated with *F. novicida* LPS for 20 h even in absence of transfection reagent (Fig. 4b). This response was significantly lower than the one observed upon direct delivery of *F. novicida* LPS into the host cytosol ($p < 0.01$, by paired *t*-test) and might be associated with a physiological LPS-internalization pathway[37] or with alternative inflammasome activation[38]. Fugene-mediated LPS transfection did not elicit robust LDH

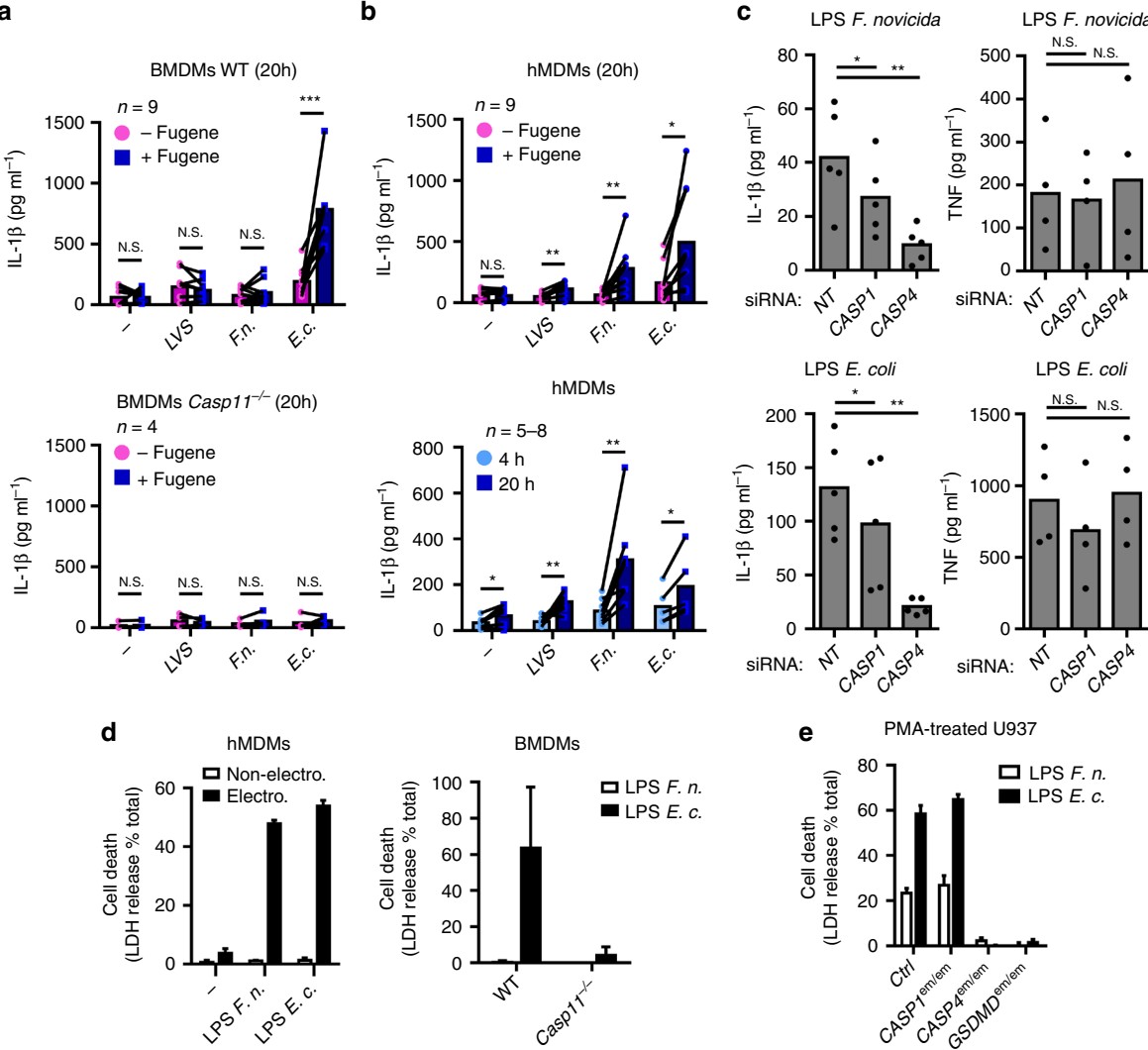

**Fig. 4** Cytosolic LPS from *F. novicida* and *F. tularensis* activates the non-canonical inflammasome in a species-dependent manner. **a** BMDMs of the indicated genotype or **b** hMDMs were primed with Pam3CSK4 and treated (Magenta, −Fugene) or transfected (Blue, +Fugene) with FugeneHD alone (−) or with 5 μg.mL⁻¹ of lipopolysaccharide (LPS) from *F. tularensis* live vaccine strain (*LVS*), *F. novicida* (*F.n.*), *E. coli* (*E.c.*). IL-1β levels were quantified by ELISA at 20 h or at the indicated time (**b**, lower panel) post-transfection. **c** hMDMs were transfected with non-targeting (NT) siRNA or the indicated siRNA, primed with Pam3CSK4, and transfected with 5 μg.mL⁻¹ of the indicated LPS. IL-1β and TNF levels were quantified by ELISA at 20 h post-transfection. **d** hMDMs, BMDMs and **e** U937 cell lines from the indicated genotypes were, as indicated, electroporated or not with 5 μg of LPS from *F. novicida* (*F.n.*) or *E. coli* (*E.c.*). Cell death was assessed by measuring lactate dehydrogenase (LDH) release at 1 h (hMDMs) or 4 h (U937, BMDMs) post-electroporation. **a–c** Each point shows the mean value from three technical replicates for one **a** mouse or **b**, **c** healthy donor. The bar shows the mean for all donors (*n* ≥ 4). **a**, **b** the lines show the pairing of the values for a single mouse or healthy donor with or without LPS transfection (+ or −FugeneHD) or (**b**, lower panel) at 4 h and 20 h post-transfection. Two-tailed *p* values with the following nomenclature (****p* < 0.001, ***p* < 0.01, and **p* < 0.05 by paired *t*-test) are shown. N.S. not significant. **d**, **e** Mean values ± standard deviation from **d** two to **e** three independent experiments are shown

release in primary hMDMs. In contrast, electroporation of *E. coli* LPS triggered a robust and fast LDH release in both hMDMs and murine BMDMs (in a caspase-11-dependent manner) (Fig. 4d). Importantly, electroporation of *F. novicida* LPS in hMDMs triggered LDH release, whereas it failed to do so in murine BMDMs (Fig. 4d). Knockdown of caspase-4 substantially reduced LDH release upon electroporation of *F. novicida* LPS in hMDMs (Supplementary Fig. 9b). The difference between murine and human cells to *F. novicida* LPS was further confirmed using U937 cell lines. Indeed, electroporation of *F. novicida* cytosolic LPS into U937 cytosol triggered caspase-4/gasdermin D-dependent LDH release (Fig. 4e), whereas murine BMDMs were highly resistant to *F.*

*novicida* LPS-mediated cell death (Fig. 4d; Supplementary Fig. 10).

Caspase-4-mediated responses to *Francisella* LPS were readily observable in human cells. Yet, the responses elicited by *Francisella* LPS were less potent and less rapid than the ones elicited by *E. coli* LPS (Fig. 4b; Supplementary Fig. 10). A dose-dependent experiment indicated that *F. novicida* LPS was 10-fold less potent than *E. coli* LPS to trigger U937 cell death (Supplementary Fig. 10). Altogether, our result demonstrates that human caspase-4 has evolved to detect *Francisella* LPS although with a lower sensitivity than *E. coli* LPS. Importantly, the function of caspase-4 during *F. novicida* infection validates

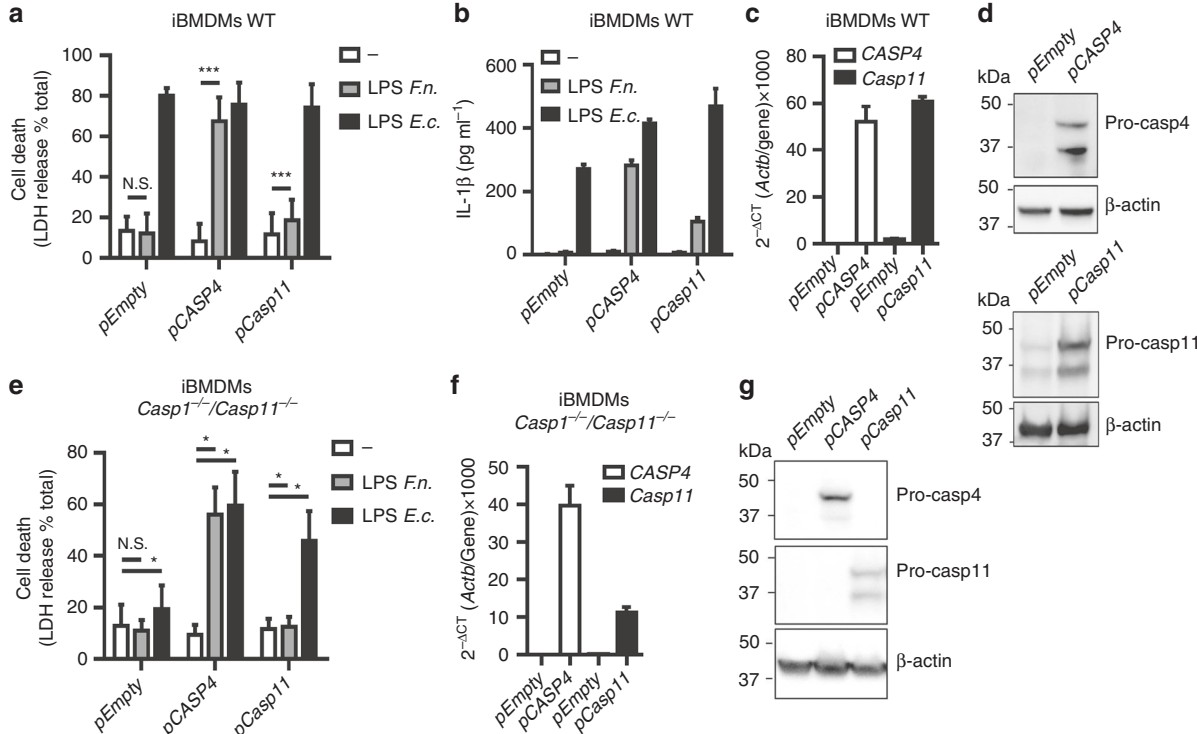

**Fig. 5** The human-specific sensing of *F. novicida* LPS is due to intrinsic properties of caspase-4. **a**, **b** WT or **e** *Casp1[−/−]/Casp11[−/−]* iBMDMs expressing the indicated constructs were electroporated with buffer only (−) or with 5 μg of *F. novicida* (*F.n.*) or *E. coli* lipopolysaccharide (LPS) (*E.c.*) LPS. **a**, **e** Cell death and **b** IL-1β levels were quantified at 4 h post-electroporation by ELISA and lactate dehydrogenase (LDH) assay, respectively. **c**, **f** The levels of the indicated transcripts in transduced immortalized bone marrow-derived macrophages (iBMDM) were assessed by qRT-PCR. **d**, **g** Ectopic expression of caspase-4 and -11 were verified by Western blot. **a**, **b**, **e** Mean values ± SD from **a** three and **e** five independent experiments, or from **b** three technical replicates from one experiment representative of three independent experiments and two-tailed *p* values with the following nomenclature (***$p < 0.001$ and *$p < 0.05$ by paired *t*-test) are shown. N.S. not significant. **c**, **f** Mean values ± standard deviation from three technical replicates are shown

the relevance of the caspase-4 response observed after direct delivery of *Francisella* LPS into the cytosol of human macrophages.

**Caspase-4 is intrinsically able to detect under-acylated LPS**. The ability of human macrophages to sense *F. novicida* LPS might be due either to an intrinsic broader sensitivity of human caspase-4 than murine caspase-11 or to the presence of human-specific co-factors licensing *F. novicida* LPS detection by caspase-4 in the cytosol of macrophages. To test these different possibilities, we ectopically expressed caspase-4 or caspase-11 in murine immortalized WT BMDMs. Expression of caspase-4 rendered murine macrophages highly sensitive to *F. novicida* LPS-mediated death (Fig. 5a). Similarly, expression of caspase-4 rendered iBMDMs able to secrete IL-1β in response to *F. novicida* LPS electroporation (Fig. 5b). In contrast, overexpression of caspase-11 (60-fold increase compared to the level of endogenous transcript, Fig. 5c, d) barely increased the sensitivity of murine macrophages to *F. novicida* LPS-mediated death and their ability to secrete IL-1β. These results obtained in WT iBMDMs were validated in *Casp1[−/−]/Casp11[−/−]* iBMDMs (Fig. 5e–g) in which we demonstrated that caspase-4 expression rescued the cell death response to both *F. novicida* and *E. coli* LPS. In contrast, ectopic expression of caspase-11 rescued the cell death response of *Casp1[−/−]/Casp11[−/−]* iBMDMs only upon *E. coli* LPS electroporation. These results demonstrated that caspase-4, in contrast to caspase-11, has the inherent ability to sense *F. novicida* LPS.

To assess the functionality of caspase-4 to directly sense under-acylated LPS, we transfected hMDMs with various under-acylated

LPS and lipid A. *F. novicida* lipid A is tetra-acylated[20]. Transfection of tetra-acylated lipid A (lipid IVa) into hMDMs elicited IL-1β release (Fig. 6a), whereas we could not detect any caspase-11-dependent IL-1β release in murine BMDMs upon lipid IVa transfection (Fig. 6b). These experiments were performed using liposome (FugeneHD)-mediated transfection as electroporation is not highly efficient to deliver uncharged molecules such as lipid A or lipid IVa into the host cytosol.

A clinical isolate from *Bacteroides vulgatus*, a gut microbe associated with intestinal inflammation and insulin resistance[39, 40], expresses a LPS composed in majority of a monophosphoryl penta-acylated lipid A[41]. As previously observed with *F. novicida* LPS, *B. vulgatus* LPS did not elicit caspase-11-dependent IL-1β release upon transfection in BMDMs (Fig. 6b). Yet, transfection of *B. vulgatus* LPS in hMDMs induced a robust IL-1β release (Fig. 6a) further strengthening the different specificities of human and murine macrophages in sensing LPS. Under-acylated LPS from a *P. aeruginosa* cystic fibrosis isolate and from *Rhodobacter sphaeroides* elicited heterogeneous IL-1β responses in different human individuals (Supplementary Fig. 11). This result suggests that inter-individual differences in LPS sensing specificities might exist in addition to the inter-species differences demonstrated by the present study.

**GBPs promote activation of caspase-4**. Along the course of our experiments, we noticed that IFN-γ priming enhanced inflammasome activation in response to *F. novicida* infection (Supplementary Fig. 12) or to *Francisella* LPS transfection (Supplementary Fig. 13) in hMDMs. We observed a substantial

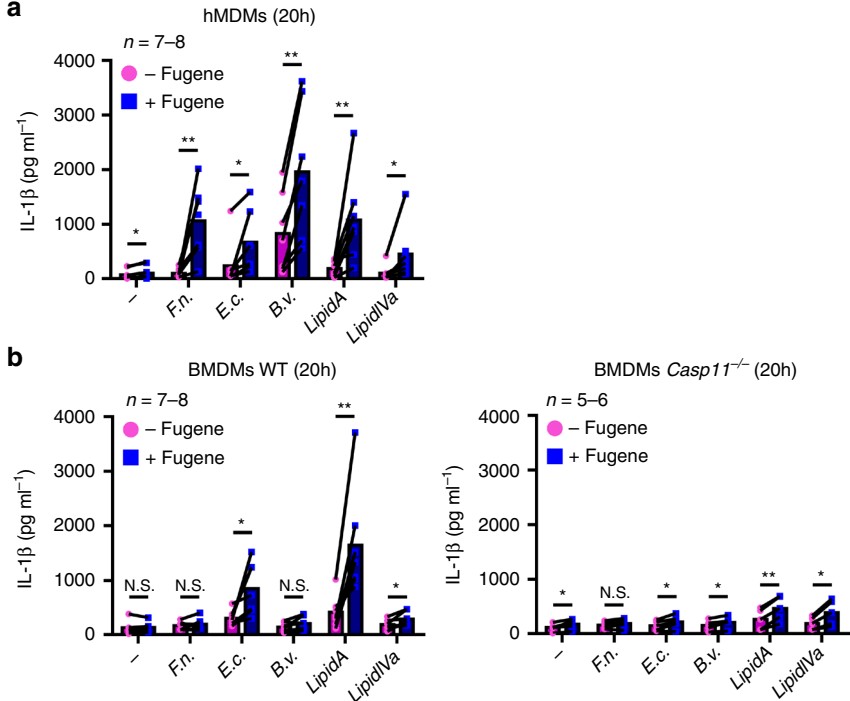

**Fig. 6** Macrophages respond to various under-acylated LPS and to synthetic tetra-acylated lipid A in a species-specific manner. **a** hMDMs and **b** BMDMs from the indicated genotype were primed overnight with IFN-γ followed by priming with Pam3CSK4 and treatment or transfection with buffer (−) or with 5 μg.mL⁻¹ of lipopolysaccharide (LPS) from *F. novicida* (*F.n.*), *E. coli* (*E.c.*), *B. vulgatus* (*B.v.*), or with 5 μg.mL⁻¹ of synthetic hexa-acylated (lipid A), tetra-acylated (lipid IVa) lipid A. IL-1β secretion was measured by ELISA at 20 h post-transfection. Each point shows the mean value from three technical replicates for one mouse or one healthy donor. The bar shows the mean for all mice ($n \geq 7$) or donors ($n \geq 7$). The lines show the pairing of the values for a single mouse or healthy donor with or without LPS transfection (+ or − FugeneHD). Two-tailed $p$ values with the following nomenclature (**$p < 0.01$ and *$p < 0.05$ by paired $t$-test) are shown, N.S. not significant

impact of IFN-γ on IL-1β release upon *E. coli* LPS transfection only at early time post-transfection. This result suggested that IFN-inducible genes might increase the cytosolic innate immune responses to under-acylated LPS in hMDMs. The guanylate-binding proteins Gbp2 and Gbp5 are IFN-inducible proteins triggering *F. novicida* lysis in BMDMs[25, 27]. Furthermore, Gbps potentiate caspase-11 activation in response to *Salmonella*, *Legionella*, and *Chlamydia* infections[11, 42, 43]. Owing to differences in *F. novicida* inflammasome sensing, as well as in GBP proteins between mice and humans[28, 44] and given the current lack of knowledge on the function of human GBPs in regulating inflammasome activation, we wondered whether GBPs could participate in the sensing of under-acylated LPS in hMDMs. Seven *GBP* genes are present in humans. Five of them (*GBP1 to 5*) were expressed in hMDMs and their expressions were strongly induced upon infection with *F. novicida* (Fig. 7a). Infection with the ΔFPI *F. novicida* mutant led to a lower induction likely related to the inability of this vacuolar mutant to trigger the cGAS pathway and type I IFN induction (Supplementary Fig. 14)[45]. IFN-γ priming increased *GBP1–5* expression and potentiated *F. novicida* infection to induce expression of most *GBPs* in hMDMs (Supplementary Fig. 15). To determine the impact of GBPs on the non-canonical inflammasome activation, we evaluated the effect of *GBP1–5* knockdown (Supplementary Fig. 3b) in hMDMs infected with *F. novicida*. *GBP2* knockdown led to a robust reduction in IL-1β, IL-18, and IL-1α levels, and decreased LDH release in the supernatant of *F. novicida*-infected hMDMs (Fig. 7b, c). Knockdown of *GBP1*, *3*, and *4* significantly reduced LDH release (Fig. 7c, $p < 0.001$, $p < 0.01$, $p < 0.05$, respectively by paired $t$-tests) while affecting in a heterogeneous manner caspase-

4-dependent cytokines release (Fig. 7b). In contrast, we did not observe any function for GBP5 in *F. novicida*-mediated cytosolic responses in hMDMs. Furthermore, none of the *GBPs* knockdown significantly modified TNF release (Fig. 7b, $p > 0.05$, by $t$-test) demonstrating that GBPs specifically control the cytosolic immune responses. These results strongly suggested that caspase-4 activation in *F. novicida*-infected hMDMs involves different GBPs. As we observed the most robust alteration of the caspase-4-dependent responses with *GBP2* knockdown, we focused on this particular GBP.

GBP2 was recruited onto WT *F. novicida* in hMDMs, whereas we did not detect any colocalisation with the ΔFPI mutant (Fig. 8a). The specificity of the GBP2 immunodetection was verified using CRISPR-Cas9 in the U937 macrophage cell line (Supplementary Fig. 16). In agreement with the decreased cell death and cytokine release (Fig. 7b, c), *GBP2* knockdown decreased caspase-1 and IL-1β processing as determined by immunoblot analysis (Fig. 8b). This result demonstrated that human GBP2 substantially contribute to the non-canonical inflammasome activation during *F. novicida* infection. The function of murine Gbps in promoting LPS sensing by caspase-11 during Gram-negative infection is well established[42, 43]. Nevertheless, the mechanisms by which GBPs promote caspase-11 activation upon direct delivery of LPS into the host cytosol is controversial[42, 43]. To assess whether GBP2 could promote IL-1β release upon direct LPS transfection in hMDMs, we performed *GBP2* knockdown in hMDMs followed by transfection with *F. novicida* LPS or *E. coli* LPS. Intriguingly, *GBP2* knockdown inhibited effectively the secretion of IL-1β induced by *F. novicida* LPS but did not impact the level of secreted IL-1β after *E. coli* LPS

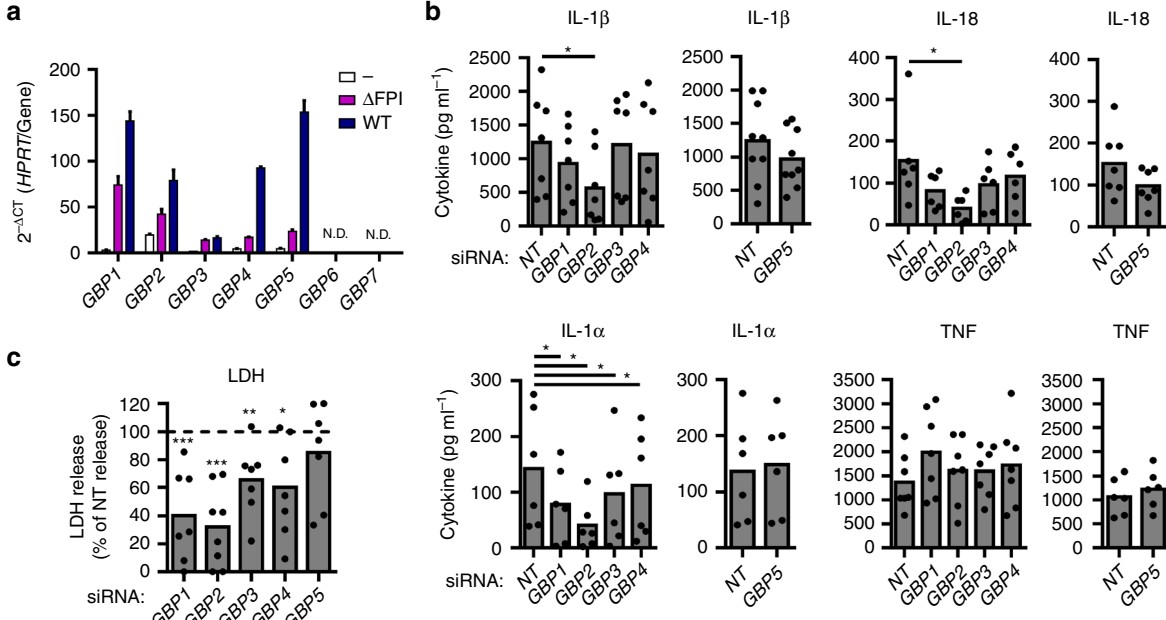

**Fig. 7** Multiple GBPs promote non-canonical inflammasome activation in hMDMs infected with *F. novicida*. **a** Expression profile of *GBPs* by qRT-PCR in hMDMs infected with wild-type *F. novicida* or the ΔFPI mutant at a multiplicity of infection (MOI) of 10 for 8 h. **b, c** hMDMs were transfected with non-targeting siRNA (NT) or with the indicated siRNA and infected with *F. novicida* at a MOI of 10 for 6 h (IL-1β, TNF, LDH) or 8 h (IL-18 and IL-1α). **b** IL-1β, IL-18, IL-1α, and TNF levels in the supernatant were quantified by ELISA. **c** Cell death was assessed by measuring LDH release. **a** Mean values ± standard deviation from three independent experiments are shown. **b, c** Each point shows the mean value from three technical replicates for one healthy donor. The bar shows the mean for all donors ($n \geq 6$). **c** For each healthy donor, the mean value from cells transfected with NT siRNA was used as a reference value set to 100%. Two-tailed $p$ values with the following nomenclature (***$p < 0.001$, **$p < 0.01$, and *$p < 0.05$ by paired $t$-test) are shown

transfection (Fig. 8c). In contrast, *NLRP3* knockdown significantly reduced IL-1β level after transfection of *F. novicida* and *E. coli* LPS (Fig. 8c, $p < 0.01$ and $p < 0.05$, respectively, by paired $t$-tests). As expected, neither *GBP2* nor *NLRP3* knockdown reduced TNF release (Fig. 8c). Altogether, our results demonstrate that GBP2 promotes the efficient sensing of *F. novicida* and tetra-acylated LPS by caspase-4 in human macrophages. In contrast, in our experimental settings, GBP2 did not promote caspase-4 activation in response to hexa-acylated LPS suggesting that GBP2 is a key co-factor to sensitize human caspase-4 to non-canonical (i.e., non hexa-acylated) LPS.

## Discussion

Here, we identified that the physiological response of human macrophages to cytosolic *F. novicida* infection relies on caspase-4, a LPS cytosolic sensor. Our results are based on siRNA-mediated gene expression knockdown in primary hMDMs which typically leads to 40 to 70% reduction in the corresponding transcript level (Supplementary Fig. 3b). These knockdown levels are in agreement with other studies using similar primary human cells[14, 37]. A limitation of this approach is that the total amount of a signaling protein (e.g., pro-caspase-1) might not be the rate-limiting factor for inflammasome activation, cell death, and cytokine production. Yet, all the results obtained in primary hMDMs were validated using CRISPR/Cas9 technology in the U937 cell line thereby supporting the siRNA results.

In lines with a previous study[21] and with studies demonstrating a predominant function of Aim2 in sensing *F. novicida* in murine phagocytes[46–48], we were unable to detect activation of murine caspase-11 and IL-1β release upon transfection of *Francisella* LPS in murine macrophages. Surprisingly, we observed that transfection of *F. novicida* or *F. tularensis* LPS led to caspase-4-

dependent IL-1β release and cell death in human macrophages. *Francisella* LPS is tetra-acylated. Caspase-11 and its human ortholog caspase-4 bind both hexa-acylated and tetra-acylated lipid As[49]. Yet, purified caspase-11 and caspase-4 proteins were first reported to oligomerize and display a catalytic activity only upon hexa-acylated LPS binding[49] thus explaining the lack of caspase-11 activation upon transfection of tetra-acylated LPS. Caspase monomers display a LPS-dependent catalytic activity although it is lower than the one of caspase oligomers[50], which suggests that oligomerization is not fully required for caspase-11 activity. Furthermore, oligomerization of caspase-4 in response to under-acylated LPS might be different in vitro and in cellulo possibly explaining the discrepancies between our study and a previous study based on recombinant caspase-4[49]. Indeed, in cellulo oligomerization of caspase-4 might be triggered by the dense packing of LPS inside bacterial membrane, within liposome or in high molecular mass aggregates[51]. Alternatively, a human-specific co-factor might favor in cellulo caspase-4 oligomerization in response to under-acylated LPS, although our results suggest that caspase-4 has an intrinsic ability to trigger inflammasome activation in response to *F. novicida* LPS. Although our results clearly demonstrate that human macrophages detect under-acylated LPS, *F. novicida* LPS is ≈10 times less potent than hexa-acylated *E. coli* LPS to elicit caspase-4 activation upon transfection (possibly due to a defect in caspase oligomerization). The caspase-4-dependent response to *F. novicida* LPS is a physiological response as it fully controls IL-1β release and cell death upon infection of hMDMs.

Under-acylated LPS escapes caspase-11-detection (ref. [21], our data). The difference in sensitivity to *F. novicida* LPS might be due to differences between the CARD domains of caspase-11 and caspase-4, which share 51% identity. Interestingly, the CARD domain of caspase-5 is even more divergent from the caspase-11

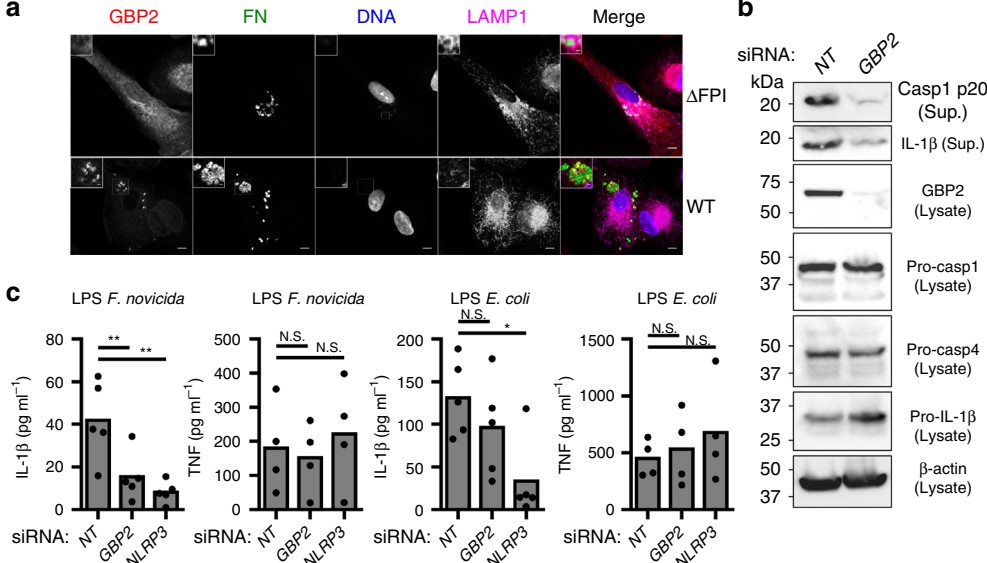

**Fig. 8** GBP2 contributes to caspase-4 responses to *F. novicida* LPS. **a** hMDMs infected with the wild-type *F. novicida* strain or the ΔFPI mutant at a MOI of 10 for 9 h were immunostained for *F. novicida* (FN, green), GBP2 (red), LAMP1 (magenta), and stained with DAPI (blue). Scale bars 5 μm and 1 μm in main images and insets, respectively. **b, c** hMDMs were transfected with non-targeting siRNA (NT) or with the indicated siRNA. **b** Caspase-1 and IL-1β processing were evaluated in the supernatant (Sup.) of hMDMs infected with *F. novicida* at a MOI of 10 for 8 h, one experiment representative of three experiments is shown. **c** hMDMs were primed with Pam3CSK4 and transfected with 5 μg.mL⁻¹ of the indicated LPS. IL-1β and TNF levels in the supernatant were quantified by ELISA at 20 h post-transfection. Each point shows the mean value from three technical replicates for one healthy donor, the bar shows the mean for all donors ($n \geq 4$). Two-tailed *p* values with the following nomenclature (\*\**p* < 0.01 and \**p* < 0.05 by paired *t*-test) are shown. N.S. not significant

CARD domain (39 and 56% identity with caspase-11 and caspase-4 CARD domains, respectively). This divergence suggests that caspase-5 CARD domain might display different specificities in lipid As binding/response. We were unable to assess the function of endogenous caspase-5 in our experimental settings since, as previously described[49], caspase-5 expression was undetectable in U937 cells at homeostasis and as the strong induction of caspase-5 expression in hMDMs upon infection (Supplementary Fig. 5a) precluded efficient knockdown of caspase-5 at the protein level using siRNA. Yet, ectopic expression of caspase-5 partially restored the ability of *CASP4^em/em^* U937 cells to activate the inflammasome in response to *E. coli* LPS transfection but not in response to *F. novicida* infection or to *F. novicida* LPS transfection (Supplementary Fig. 5) suggesting that caspase-5 by itself cannot trigger inflammasome activation in response to *F. novicida* LPS.

Interestingly, the kinetics of *F. novicida* escape into the host cytosol (hence the kinetics of cytosolic LPS exposure) are much faster (30 min to 2 h)[52] than the kinetics of caspase-4-mediated responses. This temporal segregation between the LPS presence in the cytosol and caspase-4 responses suggests either the need for a licensing/priming signal for caspase-4 activation or the need for an inducible co-factor to extract LPS for its presentation to caspase-4. Importantly, upon transfection of hMDMs with *F. novicida* LPS, we observed that IL-1β levels in the supernatant increased from 4 to 20 h post-transfection. This phenotype was less consistently observed upon *E. coli* LPS transfection (Fig. 4b), which leads to a fast IL-1β release in agreement with the described all-or-none inflammasome responses[53]. The kinetics of IL-1β secretion upon *F. novicida* LPS transfection may indicate that caspase-4 activation in response to under-acylated LPS could lead under certain conditions to the gradual release of IL-1β.

IL-1α release was strongly inhibited by NLRP3 knockdown, whereas the latter had not impact on cell death. This result is at odds with the current model derived mostly from murine studies, which indicate that the release of IL-1α upon caspase-11 activation is directly associated with necrotic cell death[5, 33]. In agreement with a previous study by Shin and collaborators[34], we did not observe any impact of *CASP1* knockdown on the release of IL-1α. Although a residual caspase-1 activity might explain this apparent lack of role, further studies are required to investigate a potential caspase-1-independent function of NLRP3 in promoting IL-1α release in human cells.

Our results demonstrate that the ability of hMDMs to detect and respond to *F. novicida* LPS is due to an intrinsic ability of caspase-4 to sense *F. novicida* LPS. In addition, host IFN-inducible factors cooperate with caspase-4 to promote the response to under-acylated LPS. GBPs are emerging as co-factors for inflammasome activation[44]. GBPs likely favor LPS extraction from *F. novicida* outer membrane and possibly from liposome or host-cell membrane upon direct transfection. Indeed, we observed that GBP2 closely co-localized with intracellular *F. novicida* and intracellular *F. novicida* remnants. This colocalization was observed with *F. novicida* expressing the FPI-encoded Type VI secretion system (T6SS) but not with the ΔFPI mutant. GBPs are recruited to pathogen-containing vacuole furnished with T3SS or T4SS[54]. Although our observations extend the correlation between GBPs recruitment and bacterial secretion systems to the T6SS, it is still unclear whether GBPs recruitment occurs directly onto cytosolic *F. novicida* or early during infection onto *F. novicida*-containing vacuoles. Interestingly, most of the expressed GBPs (with the exception of GBP5) contributed to caspase-4-mediated response. Gbps attack *Toxoplasma gondii* as supramolecular heteromers in murine macrophages[28]. The

formation of such complexes in hMDMs exposed to *F. novicida* may explain the contribution of multiple GBPs to inflammasome activation although we cannot exclude that each individual GBP has unique functions to facilitate caspase-4 activation. Remarkably, *GBP2* knockdown has the most dramatic effect on all the caspase-4-dependent phenotypes suggesting a predominant contribution of this specific GBP in *F. novicida*-mediated responses. GBP2 may thus act as the central GBP driving recruitment of GBP2 homomultimers and heteromultimers to *F. novicida* to promote LPS release and/or LPS-mediated caspase-4 activation. Interestingly, we did not observe a requirement for GBP2 upon *E. coli* LPS transfection suggesting that GBP2 is a co-factor most importantly required to trigger caspase-4 activation in response to under-acylated LPS. The ability of GBPs to oligomerize might compensate the inability of caspase-4 to oligomerize upon binding under-acylated LPS[49].

We demonstrated here that AIM2 controls only a minor way the hMDMs response to cytosolic *F. novicida*. At odds with the murine response to *F. novicida*, Aim2 does not substantially contribute to the innate immune response of murine BMDMs infected with *F. tularensis*[35]. The lack of AIM2 activation may thus result from both bacterial and host factors. Transfection of poly(dA:dT) in *F. novicida*-infected hMDMs triggered IL-1β release and cell death suggesting that *F. novicida* does not inhibit cytosolic DNA-mediated inflammasome activation (Supplementary Fig. 17). One obvious host factor lacking in human cells is the IRG defense system[29]. This system includes in mice IrgB10, a protein involved in *F. novicida* lysis, but also numerous other Irg proteins, which act in a cooperative manner to target intracellular pathogens[55]. Absence of Irg in the cytosol of hMDMs might thus impair the release of *F. novicida* genomic DNA into the host cytosol and the subsequent AIM2 inflammasome activation. Alternatively, the relative abundance of the different inflammasome sensors that differs greatly between murine an human macrophages (Supplementary Fig. 18) might explain the different contribution of the AIM2 and NLRP3 inflammasomes in the different species. Importantly, our result is consistent with the observation that in human myeloid cells, the cGAS-STING pathway is predominant over the AIM2 pathway to control inflammasome activation upon cytosolic DNA sensing[56].

Our study indicates that human macrophages are more responsive than murine macrophages to tetra-acylated and penta-acylated LPS. Although the human responses to under-acylated lipid A and LPS are lower than the responses to hexa-acylated LPS/lipid A, they are likely to be important in certain infectious settings as exemplified here with *F. novicida*. This result is consistent with previous observations indicating a higher susceptibility of humans to LPS-induced septic shock compared to rodents[57, 58]. Most bacterial pathogens display a modified LPS to escape innate immune sensing and resist antimicrobial defenses. The divergent responses to different LPS/ lipid As between human and mouse emphasize the need for an increased knowledge on LPS sensing pathways and lipid As-binding specificities in human cells and for better preclinical animal or cellular models to develop therapeutic strategies in the treatment of sepsis and septic shock.

## Methods

**Ethics statement**. Blood was obtained from healthy donors from the Etablissement Français du Sang Auvergne Rhône Alpes, France under the convention EFS 16–2066. Informed consent was obtained from all subjects in accordance with the Declaration of Helsinki. Ethical approval was obtained from Comité de Protection des Personnes SUD-EST IV (L16-189). The protocol to obtain murine bone marrow-derived macrophages was reviewed and approved by the animal ethics committee (CECCAPP, Lyon) of the University of Lyon, France under the protocol

number #ENS_2012_061, and in strict accordance with the European regulations (#2010/63/UE from 2010/09/22) and the French laws ("Décret n 2013-118 du 1er février 2013 relatif à la protection des animaux utilisés à des fins scientifiques" and "Arrêté ministériel du 1er février 2013 relatif à l'évaluation éthique et à l'autorisation des projets impliquant l'utilisation d'animaux dans des procédures expérimentales").

**Cell culture**. Primary human peripheral blood mononuclear cells (PBMCs) were isolated by Ficoll-Hypaque (Eurobio, #CMSMS201-01) density gradient centrifugation. CD14⁺ magnetic isolation kit (Miltenyi Biotec, #130-050-201) was used to isolate monocytes following the manufacturer's instructions. Monocytes and the human myeloid cell line U937 (obtained from the biological resource center, Cellulonet) were grown in RPMI 1640 medium with glutaMAX-I (#61870-010) supplemented with 10% (vol/vol) FBS, 2 mM L-glutamine, 100 IU.mL⁻¹ penicillin, 100 μg.mL⁻¹ streptomycin (all from ThermoFischer Scientific). Monocytes were exposed to 100 ng.mL⁻¹ macrophage colony-stimulating factor (M-CSF; ImmunoTools) and U937 cells to 100 ng.mL⁻¹ of phorbol 12-myristate 13-acetate (PMA; InvivoGen). Bone marrow-derived macrophages (BMDMs) from WT (Charles River, stock 000664) or Casp11⁻/⁻[59] C57BL/6 J mice were obtained and cultured as previously described[60]. Mice were bred in a specific pathogen-free animal facility (PBES, Lyon) in ventilated racks. Six to twelve-weeks-old males and females were euthanized by cervical dislocation before collection of bones. C57BL/6 mice-derived immortalized BMDMs (iBMDMs), Casp1⁻/⁻/Casp11⁻/⁻ iBMDMs (obtained from D. Monack) and 293 T cells (obtained from the biological resource center, Cellulonet), all tested mycoplasma-free were grown in Dulbecco's Modified Eagle's medium (DMEM)

**Macrophage stimulation**. To induce canonical inflammasome cells were primed with 1 μg.mL⁻¹ Pam3CSK4 (InvivoGen) for 4 h followed by treatment with 125 ng.mL⁻¹ TcdB (Clostridium difficile toxin B, Abcam), 5 μM Nigericin (InvivoGen) or transfection with 0.1 μg of poly(dA:dT) (InvivoGen) using lipofectamine® 2000 transfection reagent (ThermoFischer Scientific) for 2 h (Nigericin) or 3 h (TcdB and poly(dA:dT)).

**Bacterial strains and macrophage infection**. *F. novicida* strain Utah (U112) and its isogenic mutant lacking the whole FPI[61] were grown in tryptic soy broth (Conda) supplemented with 0.1% (w/v) cysteine. U937 cells, hMDMs, and BMDMs were plated at a concentration of 3 to 5 × 10⁴ cells in 96-well plates or 3 to 5 × 10⁵ in 24-well plates. One day before infection, hMDMs were gently detached and plated in media without antibiotics supplemented with 100 ng.mL⁻¹ M-CSF. U937 cells were treated 24 h with PMA before overnight stimulation with 100 U.mL⁻¹ IFN-γ (Immunotools) followed by infection. Bacteria were added onto macrophages at the indicated multiplicity of infection (MOI). The plates were centrifuged for 15 min at 1000×g and incubated for 1 h at 37 °C. Next, cells were washed and fresh medium with 10 μg.ml⁻¹ gentamycin (ThermoFisher Scientific) was added before incubation for the indicated time. When indicated, hMDMs were pretreated for 30 min before infection with the caspase inhibitors z-YVAD-FMK or z-LEVD-FMK (Enzo LifeSciences) at the indicated concentrations, inhibitors were maintained during infection. When indicated, infected hMDMs were transfected at 2 h post-invasion with 0.1 to 0.5 μg of poly(dA:dT) for 3 h.

**CRISPR/Cas9**. CRISPR/Cas9-mediated gene invalidations are identified as 'em' for endonuclease-mediated. The cGAS^em/em THP-1 cell line (obtained from V. Hornung) was previously described[62] and tested mycoplasma-free. GBP2^em/em, NLRP3^em/em, AIM2^em/em and MEFV^em/em U937 cell lines were generated as followed. gRNA directed against GBP2 and NLRP3 were cloned into the pSpCas9 (BB)-2A-GFP (PX458) vector (from Feng Zhang; Addgene plasmid # 48138) following an adapted protocol from Zhang laboratory[63] using specific gRNA (Supplementary Table 1). gRNAs directed against AIM2 and MEFV were cloned into a px330A-1×2 derivative (from Takashi Yamamoto; Addgene plasmid # 58766) modified to express GFP following an adapted protocol from Yamamoto laboratory[64]. To generate the knockout cell line, 10 μg of gRNA-expressing plasmid were transfected into mycoplasma-free U937 cells (obtained from the biological Resource Center-Cellulonet) using Neon® Transfection System (ThermoFischer Scientific, #MPK10096) according to manufacturer's protocol. Two days later, single clone GFP-positive cells were sorted by flow cytometry on the BD Biosciences FACSAria II. GBP2 and NLRP3 knockout clones were screened by sequencing of a PCR fragment corresponding to the genomic region flanking the targeted sequence and by Western blot. AIM2 and MEFV knockout clones were screened by gel-analysis and sequencing of a PCR fragment corresponding to the genomic region flanking the targeted sequence. Mutations of CASP1, CASP4 and GSDMD were performed in a Cas9-expressing U937 clone obtained by transduction with the plasmid LentiCas9-Blast (from Feng Zhang; Addgene plasmid # 52962) followed by blasticidin selection and clonal isolation using the limit dilution method. A clone strongly expressing Cas9 was selected based on Western blot analysis using anti-Cas9 antibody (Millipore; # MAC133; 1:1000 dilution). gRNA targeting CASP1, CASP4, and GSDMD (Supplementary Table 1) were cloned into

the pKLV-U6gRNA(BbsI)-PGKpuro2ABFP vector (from Kosuke Yusa; Addgene plasmid # 50946). For each gene, two pairs of gRNAs were used. Lentiviral particles were produced in 293T cells using pMD2.G and psPAX2 (from Didier Trono, Addgene plasmids #12259 and #12260), and pKLV-U6gRNA(BbsI)-PGKpur-o2ABFP. Cas9-expressing U937 cells were transduced by spinoculation. A polyclonal population was selected using puromycin treatment. Gene invalidation was verified by Western blotting analysis.

**Stable cell line**. *GBP2* cDNA was amplified from reverse-transcribed RNA from hMDMs. *CASP4* and *Casp11* cDNA were obtained from J. Yuan. *MEFV* cDNA was synthesized by Invitrogen. 3xFlag-*CASP4*, 3xFlag-*CASP5* and 3xFlag-*MEFV* cDNAs were obtained after cloning of *CASP4*, *CASP5*, and *MEFV* cDNAs into pEntr1a-3xFlag vector. 3xFlag-*MEFV* was cloned into pInducer21 (Addgene plasmid #46948) downstream of a doxycycline-inducible promoter. 3xFlag-*CASP4*, 3xFlag-*CASP5* were cloned into pAIP lentiviral vector (from A. Cimarelli). Lentivirus were produced and cells transduced as described above. Seventy-two hours post-transduction, stably transduced cells were selected with 2 μg.mL$^{-1}$ Puromycin (Sigma-aldrich) for 10 days.

**siRNA-mediated knockdown**. siRNA pools (ON-TARGETplus; Dharmacon) targeting *MEFV* (L-011081-00), *AIM2* (L-011951-00), *NLRP3* (L-017367-00), *GBP1* (L-005153-00), *GBP2* (L-011867-00), *GBP3* (L-031864-01), *GBP4* (L-018177-01) and *GBP5* (L-018178-01) and the non-Targeting siRNA #2 (D-001210-02-05); individual siRNA (Silencer® Select, Ambion) targeting *CASP1* (s2408 and s2407) or *CASP4* (s2413 and s2414) and the Silencer Select Negative Control No. 1 siRNA were used. hMDMs were seeded into 24- or 96-well plates at a density of $3 \times 10^5$ or $3 \times 10^4$ cells per well. siRNA complexes were prepared at a concentration of 10 nM total siRNA and transfected using lipofectamine® RNAiMAX transfection reagent (ThermoFischer Scientific, #13778100) following the manufacturer's protocol.

**Real-time PCR**. Total RNA was extracted with TRI Reagent® (Sigma-Aldrich) and reverse-transcribed with random primer combined with ImProm-II™ Reverse Transcription System (Promega, #A3800). Quantitative real-time PCR was performed using FastStart Universal SYBR Green Master Mix (Roche, #04913850001) and an Applied StepOnePlus™ Real-Time PCR Systems (ThermoFisher Scientific). Gene-specific transcript levels were normalized to the amount of human *HPRT*, human *ACTB* or mouse *Actb* transcripts. Primers sequences are indicated in the Supplementary Table 1.

**Cytokine release measurement and cell death assay**. All ELISAs with the exception of the hIL-18 and mIL-18 ELISA (MBL International antibodies D044-3, D045-6 and Thermofisher #BMS267-2, respectively) were from R&D Systems (#DY210, DY201, DY401, DY400, DY200). Quantification of cell death was performed by analysis of LDH release in the cell supernatant. Cell death was also quantified by monitoring propidium iodide (PI) incorporation. Briefly, 3 to $5 \times 10^4$ U937 cells, hMDMs or BMDMs were seeded in 0.3 cm$^2$ wells of a black 96-flat-bottom-well plate. One hour after infection, PI was added at a final concentration of 5 μg.mL$^{-1}$. Fluorescence was measured over a 24 h period on a microplate fluorimeter (Tecan).

**Immunoblot analysis**. Cells were lysed with RIPA buffer containing Mini EDTA-free Protease Inhibitor Mixture (Roche, #11836170001). Chloroform/methanol precipitation was performed to concentrate proteins from supernatant. Proteins were separated by SDS/PAGE on precast 4–15% acrylamide gels (Bio-rad, #4561084) and transferred to TransBlot® Turbo™ Midi-size PVDF membranes (Bio-rad, #1704157). Antibodies used were rat monoclonal anti-Caspase-11[59] (from J. Yuan), mouse monoclonal anti-NLRP3 (clone Cryo-2; Adipogen; 1:500 dilution), mouse monoclonal anti-GBP2 (clone 5C8; Novusbio; 1:2000 dilution), mouse monoclonal anti-PYRIN (clone C-11; Santa Cruz Biotechnology; 1:1000 dilution), mouse monoclonal anti-FLAG® (clone M2; 1:1000 dilution) and rabbit polyclonal anti-GSDMD (1:1000 dilution) purchased from Sigma-Aldrich, rabbit monoclonal anti-Caspase-1 (clone D7F10; 1:1000 dilution), rabbit polyclonal anti-Caspase-4 (1:1000 dilution), monoclonal anti-Caspase-5 (clone D3G4W; 1:1000 dilution), and rabbit monoclonal anti-IL-1β (clone D3U3E; 1:1000 dilution) purchased from Cell signaling. As loading control, cell lysates were reprobed with a mouse monoclonal antibody anti-β-actin (clone C4, Milipore; 1:5000 dilution). Full western blot images are presented in Supplementary Fig. 19. Densitometric analysis was performed using ImageJ software to compare band intensities between experimental conditions. Values calculated correspond to the ratio between the intensities of the gene of interest and actin followed by normalization to the control experimental condition (set to 1).

**Delivery of intracellular LPS and lipid A**. hMDMs and BMDMs were plated at 3 to $5 \times 10^4$ cells per well of a 96-well plate or $5 \times 10^5$ cells per well of a 24-well plate. After overnight stimulation with or without 100 U.mL$^{-1}$ human or mouse IFN-γ (ImmunoTools), cells were primed with 1 μg mL$^{-1}$ Pam3CSK4 (Invivogen) for 4 h.

Culture media were then replaced with fresh media, and cells were treated with LPS or lipid A (5 μg.mL$^{-1}$), mock-transfected with FugeneHD (Promega), or treated with a mixture of FugeneHD (0.25% (vol/vol)) and LPS or lipid A (5 μg.mL$^{-1}$) per well for 4 to 20 h. When indicated, $5 \times 10^5$ hMDMs, U937, BMDMs, or iBMDMs were electroporated with LPS using Neon® Transfection System (ThermoFischer Scientific, # MPK10096) according to manufacturer's protocol. Cell death assays were performed at 1 h post-electroporation in hMDMs and 4 h post-electroporation for U937 and BMDMs. Synthetic tetra-acylated lipid A (lipid IVa) was purchased from PeptaNova. Synthetic hexa-acylated lipid A (lipid A), LPS from *E. coli* O111:B4 L (*E.c.*) and *Rhodobacter sphaeroides* LPS (*R.s.*) were purchased from InvivoGen. LPS from *F. tularensis* LVS and *F. novicida* (*F.n.*)[20] were obtained from Wayne Conlan (National Research Council Canada). LPS from *Bacteroides vulgatus* (*B.v.*) and *Pseudomonas aeruginosa* RP73 strain, a persistent cystic fibrosis isolate (*P.a.*) were isolated using standard techniques[41, 65].

**Immunofluorescence**. Primary human macrophages and U937 were seeded on glass coverslips and were infected as described above. At the desired time points, cells were washed three times with PBS and were fixed for 15 min at 37 °C with 2% paraformaldehyde. Following fixation, coverslips were washed three times with PBS. After 20 min incubation with blocking buffer (PBS, 10% Horse serum, 3% BSA, 0.1% saponin), coverslips were stained for 20 min at room temperature with the following primary antibodies: mouse anti-human GBP2 (clone 5C8, Novusbio; 1:1000), chicken anti-*F. novicida* (from D. Monack; 1:2000), and rat anti-LAMP1 (clone 1D4B, Abcam; 1:1000) then incubated for 20 min at room temperature with the following secondary antibodies: goat anti-mouse-Alexa Fluor 488 (Sigma-Aldrich; 1:1000), goat anti-chicken-Alexa Fluor 647 (Sigma-Aldrich; 1:1000) and donkey anti-rat-Alexa Fluor 594. Following a 10 min staining with 6-diamidino-2-phenylindole (DAPI; Vector Labs), coverslips were mounted on glass slides with Mowiol® (Sigma-Aldrich) and imaged on a Zeiss LSM710.

**Statistical analysis**. Prism 6 software (GraphPad) was used for statistical analysis of data. For evaluation of the differences between treatments (e.g., mock-transfected vs. LPS transfected or NLRP3 siRNA vs NT siRNA), a two-tailed *t*-test was used. The values obtained for each individual donors were paired.

**Data availability**. The authors declare that the data supporting the findings of this study are available within the paper and its Supplementary information file.

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

## Acknowledgements

We thank A. Cimarelli (CIRI, Lyon, France), W. Conlan (NRC, Ottawa, Canada), V. Dixit (Genentech, South San Francisco, USA), V. Hornung (Gene Center Munich, Munich, Germany), D. Monack (Stanford University, Palo Alto, USA), V. Petrilli (Centre de Recherche en Cancérologie de Lyon, Lyon, France), J. Yuan (Harvard Medical School, Boston, USA), for reagents; A. Bragonzhi (HSR, Milano, Italy), J.S. Frick (University of Tübingen, Tübingen, Germany) for providing *P. aeruginosa* and *B. vulgatus* strains, F. Peri (University of Milano Bicocca, Milan, Italy) for helpful discussions. We acknowledge the contribution of SFR Biosciences (UMS3444/CNRS, US8/Inserm, ENS de Lyon, UCBL) facilities PBES, Cytometry, Cellulonet and Platim. This work was supported by

ERC starting grant 311542 to T.H. and ERC consolidator grant ERC-2013-CoG_616986 to B.F.P. This work was performed within the framework of the LABEX ECOFECT (ANR-11-LABX-0048) of Université de Lyon, within the program'Investissements d'Avenir' (ANR-11-IDEX-0007) operated by the French National Research Agency (ANR).

## Author contributions

B.L., S.B., A.Mo., and T.H. designed research. B.L., S.B., F.Ma., P.W., A.P., F.Mi., and A.Ma. performed experiments. A.Mo., F.D.L., and B.F.P. contributed reagents. B.L. and T.H. wrote the manuscript.

## Additional information

**Competing interests:** The authors declare no competing financial interests.

