## [Peer Review File · Nature Communications]

Reviewers' comments:

Reviewer #1 (TLR, PRR, inflammasome) (Remarks to the Author):

The manuscript by Lagrange, Henry, and colleagues investigates the species-specific recognition and response to LPS and to Francisella infection in human and mouse cells. They conclude the human caspase-4, can respond to a broader range of bacterial LPS structures than its murine homolog, caspase-11. In particular, they find that human caspase-4 can respond to tetra-acylated or under-acylated LPS, in contrast to murine caspase-11, which only responds to hexa-acylated LPS. These data are intriguing, given that the group of Feng Shao previously reported that under-acylated LPS can bind caspase-4, but cannot induce its oligomerization, and therefore does not induce caspase-4 activation or downstream signaling. Can the authors discuss this apparent discrepancy a bit further? The authors also suggest in the abstract that human GBPs 'synergize' with caspase-4 to promote inflammasome responses to under-acylated LPS in human cells. The data in this manuscript demonstrates that the human and mouse inflammasome systems differ with respect to their sensitivity to various forms of LPS. The authors also demonstrate a key difference in the relative contributions of AIM2 and non-canonical inflammasomes in mouse and human cells infected with Francisella. This is a novel finding and has potentially important implications for our understanding of how different inflammasomes work in human cells. The authors also perform some nice mechanistic studies to try to map the difference in human vs. mouse responses to the CARD domain of caspase-4/5. Overall the study is well done and represents a nice investigation of differences between human and mouse non-canonical inflammasome activation during Francisella infection and response to cytoplasmic LPS. However, some of the conclusions in the paper aren't fully supported by the data, and there are several aspects of the study that are a bit confusing and need clarification.

Specific Comments:

1. The authors attribute the phenotypes that they see to the knockdown of caspase-4. However, caspase-5 could also potentially play a role – the siRNA knockdown of caspase-5 seems not to be efficient or even to have worked at all (Fig 2e). It therefore seems at least possible that caspase-5 is also playing a key role here. This appears to be due to the upregulation of casp5 and its resistance to knockdown – this is mentioned by the authors in passing, but I think it merits further discussion at the very least. Another rather confusing aspect of this panel is that the amount of caspase-1 p20 in the supernatant seems to be unaffected by the casp1 siRNA knockdown, even though pro-casp1 in the lysate is reduced – can the authors clarify this? This makes me worry that this p20 band being detected is non-specific.

2. The figure legend of figure 1 states that the 'NLRP3 inflammasome is the main inflammasome sensor of Francisella in human MDMs'. However, I do not think that NLRP3 is sensing Francisella at all here. The simplest explanation for the collective findings in this manuscript is that Francisella LPS triggers caspase-4/5 non-canonical inflammasome, leading to gasdermin D cleavage and pore formation, followed by potassium efflux and NLRP3 inflammasome activation. Therefore NLRP3 is not directly sensing Francisella, but rather sensing the downstream consequence of caspase-11 activation. This is supported by the data that NLRP3 knockdown does not reduce cell death/LDH release, since this is driven by caspase-4/gsdmd cleavage. This could be tested by knockdown of gasdermin D or adding extracellular potassium to the cell culture medium. A prediction of this is that caspase-1 cleavage and IL-1b release will be abrogated in both cases.

3. The authors suggest that GBPs act synergistically with caspase-4 to promote inflammasome responses to Francisella. This statement is based on the observation that both GBP knockdown and caspase-4 knockdown result in reduced inflammasome activation. However, again I am not sure that

these data support the conclusion – for this to be considered 'synergistic' GBPs and caspase-4 would need to be acting in separate pathways. It seems much more likely that the GBPs are facilitating detection of cytosolic LPS by caspase-4, either by releasing it from the bacterial surface or from LPS-containing liposomes.

4. I agree that human MDMs do not respond to *Francisella novicida* LPS. However, it does look like there is a response to lipid IVa in human MDMs. However, it is difficult to interpret this figure because the data in figure 5 in both human in mouse MDMs has a very high scatter – it is not clear whether the differences between Fugene-transfected vs. the Fugene control are in fact significant for many of the treatments – the scatter in the data, both in mice and human MDMs, making the transfection studies a bit challenging to interpret. Additionally, some controls are missing here – we only see IL-1b as a readout, and no other inflammasome stimuli (ie NLRC4) are tested in parallel with MDMs from the same individuals or mice.

5. The experiments using iBMDMs expressing Casp4/11 constructs and the various CARD-swapping protein constructs are really nice and quite intriguing. However, it is surprising that these studies were done in iBMDMs from wild-type mice – it seems that a better way to do this experiment is in iBMDMs from caspase-11 mice – the *E. coli* LPS control isn't really a good control, since the iBMDMs containing pEmpty respond perfectly fine to *E. coli* LPS, due to the fact that there is endogenous casp11 there. To this extent, it is unclear whether the pCasp11CARD-Casp4 construct is functional at all – the prediction is that it would be able to respond to *E. coli* LPS, but we can't actually make this determination, because it is confounded by the presence of wild-type Casp11 in these BMDMs.

6. With regard to interpretation of the GBP knockdown – the knockdown data are quite convincing, but these data indicate that Gbp1, 2, 3, and 4 are all individually important, not that multiple GBPs cooperate – loss of any one GBP substantially reduced the response, suggesting that they have individual and uniquely important roles in inflammasome activation.

7. A general concern I have with the way the data are presented is that they are often graphed as percent of Non-treated – why is this done? I would recommend plotting these data as absolute values of secreted cytokine, as this provides a better sense of the actual amount of cytokine produced.

8. With regard to the microscopy in Figure 7 -- the GBP2 staining should be shown to be specific either by staining GBP-deficient cells or GBP2 siRNA knockdown cells, which the authors have demonstrated has a functional effect on inflammasome activation.

Reviewer #2 (Inflammasome, Caspase 11) (Remarks to the Author):

In the manuscript entitled "Human caspase-4, in synergy with GBPs, detects tetra-acylated LPS and cytosolic *Francisella* highlighting functional differences with murine caspase-11", Brice et al., describe a new intracellular pathway by which cytosolic *F. novicida* is recognized by the host. They report that human caspase-4, unlike murine caspase-11, is capable of recognizing *F. novicida* tetra-acylated lipid A. They show that caspase-4 recognition of *F. novicida* LPS is responsible for the activation of pyroptosis and NLRP3-dependent caspase-1 activation. The strengths of this study include the utilization of human primary macrophages and the demonstration of discrepancies in human and murine cytosolic sensing system. However, as explained below, this manuscript is preliminary and requires extensive additional studies to convincingly demonstrate this phenomenon.

Major comments

A main weakness of this paper is that it reports several observations (contradictory to current literature but interesting) without data-backed explanations. For instance, AIM2 was shown to be not activated by *F. novicida* in human cells without addressing if this is due to active inhibition of AIM2 by *F. novicida* or inaccessibility of AIM2 to *F. novicida* DNA.

Does caspase-4 directly bind to *F. novicida* tetra-acylated lipid A and undergo oligomerization? It has been shown previously that caspase-4 does not oligomerize upon binding tetra-acylated LPS and without oligomerization, caspase-4 will not be activated.

In general, tetra-acylated lipid A induced responses in human cells were minimal, which is in contrast to the responses elicited by hexa-acylated lipid A.

Knockdown experiments were lacking appropriate positive and negative controls throughout the manuscript (for example Fig 1d, e, and f; Fig 2c, d, and e). Without these controls it is difficult to attribute the observed phenotypes to the RNA silencing.

Fig 2e. It is surprising that even caspase-1 knockdown did not affect the levels of cleaved caspase-1 in the supernatants.

Did *F. novicida* LPS transfection/electroporation induce cell death and caspase-1 activation in primary hMDMs?

Fig 4a. Did caspase-4 activate caspase-1 and IL-1 β secretion?

Fig 5. Did tetra-acylated lipid A transfection induce cell death in hMDMs?

Fig 4a. Cell death induced by chimeric caspase-4 CARD fused to the caspase-11 C-terminal domain was partial compared to full length WT caspase-4.

Fig S12. It is well documented that IFN-g priming augments *E. coli* LPS and *Simonella* induced noncanonical inflammasome activation (Shi et al Nature 2014 and Aachoui, et al Science 2013). Therefore the authors' claim that IFN-g priming specifically boosts inflammasome responses to *F. novicida* LPS but not *E. coli* LPS is surprising and the data presented are not convincing.

Fig 3c. It is surprising that caspase-1 knockdown only minimally affected IL-1 β secretion in response to *F. novicida* and *E. coli*.

The absolute protein values for cytokine ELISAs (pg/ml or ng/ml) should be presented instead of % of NT control.

Minor comments

Fig 1. The knockdown of NLRP3 and AIM2 should be verified at the protein levels.

Fig. 2e. There was no knockdown of caspase-5 so they shouldn't include it in the data for fig 2c.

Fig 4b. Protein expression should be shown.

Figure citations in the text are not in chronological order.

Reviewer #3 (Inflammasome, NLRP3) (Remarks to the Author):

In this paper Brice et al. propose that caspase 4 is responsible for triggering *F. novicida*-induced pyroptosis and for activating the NLRP3 inflammasome in human macrophages. Moreover, the authors state that human GBPs synergize with caspase-4 to trigger inflammasome responses to cytosolic under-acylated LPS.

The study of caspase 4/5 is important, as non canonical inflammasome activation is still poorly defined. The putative involvement of GBPs is also interesting. However, in spite the model proposed is appealing, the study displays many weaknesses. In general, the great variability of IL-1 β secretion, coupled with the low number of donors studied in some experiments, makes it difficult to reach convincing conclusions. In addition, in most cases silencing was not efficient. Moreover, many data are not compelling, and the confusion in the supplementary figures (legends missing or messy, figures that do not correspond to the description in the text) doesn't help to follow the message of the study.

In particular:

1. A crucial point in the ms is the study of the induction of pyroptosis following *F. novicida* infection. Cell death is measured as PI fluorescence and LDH release. The LDH data however display a very high variability. For instance, in fig 1e, six experiments of AIM2 silencing are shown, three of which display an LDH release of less than 50% compared to the control, whereas in two experiments twice as much LDH compared to the control is released. The same is true for fig. 2d (see LDH release by caspase 5-silenced cells!) and Fig 6c.

2. In general, silencing is highly inefficient (see supplementary figure 2 for NLRP3, AIM2 and MEFV; supplementary fig. 15 for GBPs). Silencing of caspase 1, 4, and 5, shown in figure 2e is also non convincing.

Caspase 1 silencing gave surprising results: the intracellular pro-caspase 1 is decreased but the secreted p20 is very high. The presence of secreted p20 explain why IL-1 β secretion is only slightly decreased, but does not provide information on the role of caspase 1 compared to the other caspases.

Caspase 5 is not decreased in cas5-silenced cells: again, this data explains the irrelevant effects on cytokine secretion, but does not provide information on the role of caspase 5. Actually, the only silencing that worked was that of caspase 4, that indeed results in decreased cytokine secretion. However, this result does not rule out a role for caspase 5. Moreover, since caspase 1 and 5 have not been efficiently silenced, also the conclusion: "caspase-4 also controlled *f. novicida*-mediated hMDMs death" (p. 7) is not based on solid data.

Since specific inhibitors for caspase 1 and 4/5 exist (YVAD and zLEVD), the authors could have used them to substantiate their results that, as such, are not compelling.

3. To support the difference between caspase 11 and caspase 4, the authors show that transfection of *F. novicida* and *tularensis* LPS induces IL-1 β secretion in hMDMs (Fig. 3b) but not in BMDMs (Fig. 3a). However, the low level of secreted IL-1 β coupled to the high variability among the 4 donors results in lack of significance. In Fig. 3c, again, the low inhibition of IL-1 β secretion observed in cells silenced for caspase 1 suggests that silencing did not work.

4. Figure 5 shows IL-1 β secretion by human and mouse macrophages transfected with different LPS and lipid A. While it is clear what happens in mice (transfection of LPS *E.c.* and of Lipid A induce secretion), it is very difficult to draw conclusions from the data obtained in human macrophages.

These data are too variable, the number of donors too little (3 or 4) and, as such, these results are neither statistically nor biologically significant.

We thank all the referees for their constructive comments, which we have addressed in details below and have helped us strengthen our manuscript.

Reviewer #1:

The manuscript by Lagrange, Henry, and colleagues investigates the species-specific recognition and response to LPS and to Francisella infection in human and mouse cells. They conclude the human caspase-4, can respond to a broader range of bacterial LPS structures than its murine homolog, caspase-11. In particular, they find that human caspase-4 can respond to tetra-acylated or under-acylated LPS, in contrast to murine caspase-11, which only responds to hexa-acylated LPS. These data are intriguing, given that the group of Feng Shao previously reported that under-acylated LPS can bind caspase-4, but cannot induce its oligomerization, and therefore does not induce caspase-4 activation or downstream signaling. Can the authors discuss this apparent discrepancy a bit further? The authors also suggest in the abstract that human GBPs 'synergize' with caspase-4 to promote inflammasome responses to under-acylated LPS in human cells. The data in this manuscript demonstrates that the human and mouse inflammasome systems differ with respect to their sensitivity to various forms of LPS. The authors also demonstrate a key difference in the relative contributions of AIM2 and non-canonical inflammasomes in mouse and human cells infected with Francisella. This is a novel finding and has potentially important implications for our understanding of how different inflammasomes work in human cells. The authors also perform some nice mechanistic studies to try to map the difference in human vs. mouse responses to the CARD domain of caspase-4/5. Overall the study is well done and represents a nice investigation of differences between human and mouse non-canonical inflammasome activation during Francisella infection and response to cytoplasmic LPS. However, some of the conclusions in the paper aren't fully supported by the data, and there are several aspects of the study that are a bit confusing and need clarification.

We have addressed in details the reviewer comments below. Briefly, the apparent discrepancy with the work of Feng Shao has been discussed, we have added numerous controls and CRISPR/Cas9 to validate our siRNA experiments, have added several panels to dissect the role of gasdermin D and have edited our manuscript to analyse more rigorously than before our data.

Specific Comments:

1. The authors attribute the phenotypes that they see to the knockdown of caspase-4. However, caspase-5 could also potentially play a role – the siRNA knockdown of caspase-5 seems not to be efficient or even to have worked at all (Fig 2e). It therefore seems at least possible that caspase-5 is also playing a key role here. This appears to be due to the upregulation of casp5 and its resistance to knockdown – this is mentioned by the authors in passing, but I think it merits further discussion at the very least. Another rather confusing aspect of this panel is that the amount of caspase-1 p20 in the supernatant seems to be unaffected by the casp1 siRNA knockdown, even though pro-casp1 in the lysate is reduced – can the authors clarify

this? This makes me worry that this p20 band being detected is non-specific.

We have decided to remove the data related to caspase-5 from the main manuscript since the siRNA knockdown is inefficient at the protein level as previously described. We have no evidence for a role of caspase-5 in sensing *F. novicida* LPS neither upon ectopic expression in *Casp1/11*^{-/-} iBMM (not shown), nor upon CRISPR/Cas9 gene invalidation in U937 cells (not shown) nor upon ectopic expression in *Casp4*^{em/em} U937 cells (now presented in supplemental figure S5). All the data related to caspase-5 are now presented in supplementary Fig. 5 and the following discussion has been added in the revised manuscript.

"We were unable to assess in a robust manner the functional role of caspase-5 in our experimental settings since, as previously described³⁹, caspase-5 expression was undetectable in the U937 cell line at steady state and since the strong induction of caspase-5 expression in hMDMs upon infection (supplementary Fig. 5a) precluded efficient knockdown of caspase-5 at the protein level using siRNA. Yet, ectopic expression of Caspase-5 restored the ability of *cas4*^{em/em} U937 cells to die in response to transfection of *E. coli* LPS but not in response to *F. novicida* LPS (supplementary Fig. 5) suggesting that caspase-5 by itself cannot trigger inflammasome activation in response to cytosolic *F. novicida* LPS."

We have strengthened our conclusion that the phenotype that we describe is due to caspase-4 by demonstrating that upon electroporation *F. novicida* LPS-mediated cell death is also reduced by caspase-4 knockdown (presented in supplementary Fig. 9 in the revised manuscript) and by validating all the knockdown data with CRISPR/cas9-mediated gene invalidation (see Fig. 3f,g,h for the data related to *CASP1*-, *CASP4*- and *GSDMD*- invalidated U937 cells).

Regarding caspase-1 p20, we now provide in Fig. 2e another Western blot with a lower exposure and densitometry-based quantification demonstrating a decrease (58%) in the supernatant. We are very confident that the detected band is specific since it is not detected in U937 cells invalidated for Caspase-1 (see Fig. 3h).

We tend to see a greater reduction in pro-caspase-1 level in the cell lysate than of the level of released caspase-1 p20. This is likely due to the fact that a 60 % reduction in procaspase-1 level may not induce a 60% reduction of procaspase-1 recruited to ASC upon inflammasome activation and hence not a 60% reduction in processed caspase-1 released in the supernatant. Of note, we feel the efficacy of caspase-1 knockdown is similar to other studies in the field (see for examples Fig. 2d in the study by Sunny shin and collaborators: Casson CN et al. PNAS 2015).

We have added the following sentences in the revised manuscript:

"In agreement with a role of caspase-4 upstream of the canonical inflammasome, knockdown of caspase-4 decreased caspase-1 activation as assessed by immunoblot analysis of caspase-1 p20 subunit in the supernatant (Fig. 3e-88% reduction). As expected knockdown of caspase-4 did not affect pro-caspase-1 level in the cell lysate while knockdown of caspase-1 led to a 62% reduction of pro-caspase-1 protein level. The efficacy of caspase-1 siRNA to decrease the maturation/release of casp1 p20 in the supernatant was slightly lower (58% reduction) possibly due to the number of active ASC complex being more limiting for caspase-1 maturation/release than pro-caspase-1 level. "

2. The figure legend of figure 1 states that the 'NLRP3 inflammasome is the main inflammasome sensor of Francisella in human MDMs'. However, I do not think that NLRP3 is sensing Francisella at all here. The simplest explanation for the collective findings in this manuscript is that Francisella LPS triggers caspase-4/5 non-canonical inflammasome, leading to gasdermin D cleavage and pore formation, followed by

potassium efflux and NLRP3 inflammasome activation. Therefore NLRP3 is not directly sensing francisella, but rather sensing the downstream consequence of caspase-11 activation. This is supported by the data that NLRP3 knockdown does not reduce cell death/LDH release, since this is driven by caspase-4/gsdmd cleavage. This could be tested by knockdown of gasdermin D or adding extracellular potassium to the cell culture medium. A prediction of this is that caspase-1 cleavage and IL-1b release will be abrogated in both cases.

We do agree with the reviewer and have modified the figure legend from Figure 2, which now reads:

"NLRP3 is required for *F. novicida*-mediated inflammasome activation in the cytosol of hMDMs."

In support of the Casp4/GSDMD model, we have generated a *GSDMD*-invalidated U937 cell line and have added 3 figures panels (Fig. 3f-h) demonstrating that indeed GasderminD controls Caspase-1 and IL-1 β maturation and release, cell death but not TNF release.

The following sentence has been added in the abstract:

"Here, we demonstrate that caspase-4 drives inflammasome responses to *F. novicida* infection in human macrophages. Caspase-4 triggers *F. novicida*-mediated gasderminD-dependent pyroptosis, and activates the NLRP3 inflammasome."

3. The authors suggest that GBPs act synergistically with caspase-4 to promote inflammasome responses to Francisella. This statement is based on the observation that both GBP knockdown and caspase-4 knockdown result in reduced inflammasome activation. However, again I am not sure that these data support the conclusion – for this to be considered ‘synergistic’ GBPs and caspase-4 would need to be acting in separate pathways. It seems much more likely that the GBPs are facilitating detection of cytosolic LPS by caspase-4, either by releasing it from the bacterial surface or from LPS-containing liposomes.

We agree that GBPs are likely to act upstream of caspase-4 and since we cannot fully demonstrate synergy (which would be defined as the combined effect of GBPs and caspase-4 being greater than the sum of each effects) using knockdown, we have edited the text to limit our interpretation of the data. The title now reads:

"Human caspase-4, in a GBPs-facilitated manner, detects ..."

The hypothesis regarding the role of GBPs is stated in the discussion:

"GBPs likely favour LPS extraction from *F. novicida* outer membrane and possibly from liposome or host-cell membrane upon direct transfection."

4. I agree that human MDMs do not response to Francisella novicida LPS. However, it does look like there is a response to lipid IVa in human MDMs. However, it is difficult to interpret this figure because the data in figure 5 in both human in mouse MDMs has a very high scatter – it is not clear whether the differences between Fugene-transfected vs. the Fugene control are in fact significant for many of the treatments – the scatter in the data, both in mice and human MDMs, making the transfection studies a bit challenging to interpret. Additionally, some controls are missing here – we only see IL-1b as a readout, and no other inflammasome stimuli (ie NLRC4) are tested in parallel with MDMs from the same individuals or mice.

We believe the reviewer made a mistake while referring to human MDMs instead of murine BMDMs.

The scattering of the data is due to 1) interindividual variations in human MDMs, 2) a variability between independent experiments mostly associated with transfection. We feel it is important to show combined data from independent experiments instead of a representative experiment.

To strengthen our data, we have increased the number of individuals and present in Fig. 6a, 6b data from 7 mice and 7 human donors. We now present statistical analysis of the data. This confirms our previous results that *F. novicida* LPS is not detected in murine BMDMs. We do see a minimal (although statistically significant) response of murine BMDMs to lipid IVa, but this response is similar in *casp11*^{-/-} BMDMs. We have changed our way of representing the data to facilitate the reader/ reviewer reading of the figure.

We have not used NLRC4 stimuli to compare murine and human macrophages response since we feel transfection of *E. coli* LPS allows a direct comparison of caspase-11-mediated murine and caspase-4-mediated human responses (comparison of NLRC4-mediated responses between mice and men might be further complicated due to the multiple Naip paralogs present in mice).

Regarding IL-1 β as a single readout, the experiments presented in Fig. 6a,b have been performed using Fugene-mediated LPS/ Lipid A transfection. Using this technique, we do not observe a robust cell death. We have thus limited our analysis to IL-1 β for this panel.

On the most relevant LPS for our study (*F. novicida* and *E. coli* LPS) we now provide two other readouts: i) cell death results using electroporation of primary macrophages (presented in Fig. 4d) and ii) analysis of caspase-1 processing/ release by Western blot analysis. The Fig. 4d and the supplementary Fig. S9 demonstrate that transfection of *F. novicida* LPS triggers hMDMs cell death in a caspase-4-dependent manner. We have added the following sentences in the manuscript.

" Fugene-mediated LPS transfection did not elicit robust LDH release in primary hMDMs. In contrast, electroporation of *E. coli* LPS triggered a robust and fast LDH release in both hMDMs and murine BMDMs (in a caspase-11-dependent manner) (Fig. 4D). Importantly, electroporation of *F. novicida* LPS in hMDMs triggered LDH release while it failed to do so in murine BMDMs (Fig. 4D). Knockdown of caspase-4 substantially reduced LDH release upon electroporation of *F. novicida* LPS in hMDMs (supplementary Fig. S9b)."

5. The experiments using iBMDMs expressing Casp4/11 constructs and the various CARD-swapping protein constructs are really nice and quite intriguing. However, it is surprising that these studies were done in iBMDMs from wild-type mice – it seems that a better way to do this experiment is in iBMDMs from caspase-11 mice – the E. coli LPS control isn't really a good control, since the iBMDMs containing pEmpty respond perfectly fine to E. coli LPS, due to the fact that there is endogenous casp11 there. To this extent, it is unclear whether the pCasp11CARD-Casp4 construct is functional at all – the prediction is that it would be able to respond to E. coli LPS, but we can't actually make this determination, because it is confounded by the presence of wild-type Casp11 in these BMDMs.

We had no access to *Casp11*^{-/-} iBMDMs. We have performed the transduction in *Casp1*^{-/-}/*11*^{-/-} iBMDM to investigate cell death and have added IL-1 β data obtained in WT iBMDMs. These

data presented in Fig. 5 of the revised manuscript strengthen our conclusion that caspase-4 has an intrinsic ability to detect *F. novicida* LPS while caspase-11 does not.

The results with the chimeric proteins were not as robust and reproducible in the different assays in the two different cell lines. Particularly, we failed to observe in a reproducible manner significant cell death in *Casp1/11*^{-/-} iBMDM expressing Casp11, Casp4-CARD-Casp11-Cter and Casp11-CARD-Casp4-Cter upon *F. novicida* LPS transfection (possibly due to a low expression level of Casp4-CARD-Casp11-Cter constructs).

In WT iBMDMs, the Casp4-CARD-Casp11-Cter gave, in a very consistent manner, higher IL-1 β levels than the Casp11-CARD-Casp4-Cter. Yet, on two out of three experiments, the IL-1 β levels were much lower than upon expression of Caspase-4 and were only slightly above the levels observed upon over-expression of caspase-11.

We thus need more time and experiments to understand the mechanistic behind the intrinsic ability of caspase-4 to detect *F. novicida* LPS and be fully confident in our conclusions regarding the different caspase-4 domains. We have thus decided to present in the revised manuscript only the comparison between caspase-4 and caspase-11, which gave very consistent and robust results in up to 8 independent experiments.

6. With regard to interpretation of the GBP knockdown – the knockdown data are quite convincing, but these data indicate that Gbp1, 2, 3, and 4 are all individually important, not that multiple GBPs cooperate – loss of any one GBP substantially reduced the response, suggesting that they have individual and uniquely important roles in inflammasome activation.

We agree with the reviewer that the data do not directly demonstrate the cooperation of multiple GBPs although based on the literature ² we feel this cooperation is likely to happen. We have thus edited our manuscript (including the title from Fig. 7) to have a more rigorous interpretation of our results.

The revised manuscript reads as follows:

Result section: "These results strongly suggested that caspase-4 activation in *F. novicida*-infected hMDMs involves different GBPs "

Figure 7 title: "Multiple GBPs promote non-canonical inflammasome activation in hMDMs infected with *F. novicida*."

Discussion: "The formation of "supramolecular" complexes in hMDMs exposed to *F. novicida* may explain the contribution of multiple GBPs in non-canonical inflammasome activation although we cannot exclude that each individual GBP has unique roles to facilitate caspase-4 activation."

7. A general concern I have with the way the data are presented is that they are often graphed as percent of Non-treated – why is this done? I would recommend plotting these data as absolute values of secreted cytokine, as this provides a better sense of the actual amount of cytokine produced.

The data in the revised manuscript are now presented as absolute values of secreted cytokines. This does not affect our conclusions.

8. With regard to the microscopy in Figure 7 -- the GBP2 staining should be shown to be specific either by staining GBP-deficient cells or GBP2 siRNA knockdown cells, which the authors have demonstrated has a functional effect on inflammasome activation.

This control has been performed and is provided in supplementary Fig. 16. We provide representative Western blot analyses, images and the corresponding quantification of GBP2 immunostaining in U937 cells either WT, invalidated by Crispr/cas9 for *GBP2* and the latter cell line complemented by ectopic expression of GBP2. The following sentence is present in the revised manuscript:

"Of note, the specificity of the GBP2 immunodetection was verified using CRISPR-Cas9-mediated GBP2-invalidation in the U937 macrophage cell line (Supplementary Fig. 16)."

Response to a specific point from the general comment: The apparent discrepancy with Feng Shao work has been discussed and we feel it is mostly due to differences between the behaviour of purified caspases versus our in cellulo model. The only "in cellulo" discrepancy between our work and the work by Feng Shao lies in the response of macrophages to lipid IVa. This is likely due to the different experimental model (electroporation of U937, which is not highly efficient for uncharged molecules such as lipid A and lipid IVa versus FugeneHD-mediated transfection of primary macrophages).

In agreement with Feng Shao's work, we do see that the response to under-acylated LPS is delayed and/or lower than the response to hexa-acylated LPS. We have made this clear in the revised manuscript.

We have added the following sentences in the manuscript:

In the introduction: "While the human response to under-acylated LPS/ Lipid A was not as strong as the response to hexa-acylated LPS/ Lipid A (...)"

In the result section: "These experiments were performed using liposome (FugeneHD)-mediated transfection since electroporation is not highly efficient to deliver uncharged molecules such as Lipid A or Lipid IVa into the host cytosol."

In the discussion:

"Furthermore, oligomerization of caspase-4 in response to under-acylated LPS might be different in vitro and in cellulo possibly explaining the discrepancies between our study and a previous study based on recombinant caspase-4³⁹. Indeed, in cellulo oligomerization of caspase-4 might be triggered by the dense packing of LPS inside bacterial membrane, within liposome or in high molecular mass aggregates as recently described⁵¹. Alternatively, a human specific co-factor might favor in cellulo caspase-4 oligomerization in response to under-acylated LPS, although our results suggest that caspase-4 has an intrinsic ability to trigger inflammasome activation in response to *F. novicida* LPS."

"While the human responses to under-acylated LipidA and LPS are lower than the responses to hexa-acylated LPS/ lipidA, they are likely to be important in certain infectious settings as exemplified here with *F. novicida*."

Reviewer #2

*In the manuscript entitled "Human caspase-4, in synergy with GBPs, detects tetra-acylated LPS and cytosolic Francisella highlighting functional differences with murine caspase-11", Brice et al., describe a new intracellular pathway by which cytosolic *F. novicida* is recognized by the host. They report that human caspase-4, unlike murine caspase-11, is capable of recognizing *F. novicida* tetra-acylated lipid A. They show that caspase-4 recognition of *F. novicida* LPS is responsible for the activation of pyroptosis and NLRP3-dependent caspase-1 activation. The strengths of this study include the utilization of human primary macrophages and the demonstration of discrepancies in human and murine cytosolic sensing system. However, as explained below, this manuscript is preliminary and requires extensive additional studies to convincingly demonstrate this phenomenon.*

Major comments

*1-A main weakness of this paper is that it reports several observations (contradictory to current literature but interesting) without data-backed explanations. For instance, AIM2 was shown to be not activated by *F. novicida* in human cells without addressing if this is due to active inhibition of AIM2 by *F. novicida* or inaccessibility of AIM2 to *F. novicida* DNA.*

As previously mentioned in the manuscript, the lack of AIM2 activation in human macrophages infected with *F. novicida* is in line with the current literature although contradictory to the data obtained in mice. The lack of AIM2 recognition is not the major focus of the current manuscript, which focuses on caspase-4 and we have thus not extensively investigated the underlying mechanism.

We do not see inhibition of poly(dA:dT)-mediated AIM2 inflammasome in *F. novicida*-infected hMDMs as seen by monitoring IL-1 β /LDH release in *F. novicida*-infected hMDMs following poly(dA:dT) transfection. Our data thus do not support an active inhibition of AIM2 inflammasome by *F. novicida*. We now provide the corresponding data in supplemental figure S17. Furthermore, we discuss two alternative hypotheses i-lack of Irgb10-mediated lysis and ii-differences due to the difference in AIM2/NLRP3 expression levels in murine and human macrophages (see supplementary Fig. 18 in the revised manuscript) and have added the following discussion in the revised manuscript.

"Transfection of poly(dA:dT) in *F. novicida*-infected hMDMs triggered IL-1 β release and cell death suggesting that *F. novicida* does not inhibit AIM2 inflammasome (Supplementary Fig. S17). One obvious host factor lacking in human cells is the IRG defence system²⁹. This system includes in mice IrgB10, a protein involved in *F. novicida* lysis, but also numerous other Irg proteins, which act in a cooperative manner to target intracellular pathogens⁵⁵. Absence of Irg in the cytosol of hMDMs might thus impair the release of *F. novicida* genomic DNA into the host cytosol and the subsequent AIM2 inflammasome activation. Alternatively, the relative abundance of the different inflammasome sensors that differs greatly between murine and human macrophages (Supplementary Fig. S18) might explain the different contribution of the AIM2 and NLRP3 inflammasomes in the different species."

*2-Does caspase-4 directly bind to *F. novicida* tetra-acylated lipid A and undergo oligomerization? It has been shown previously that caspase-4 does not oligomerize upon binding tetra-acylated LPS and without oligomerization, caspase-4 will not be*

activated.

Oligomerization of caspase-4 has been monitored only in vitro using recombinant proteins produced in insect cells. We have tried to monitor in cellulo caspase-4 oligomerization using a cross-linking agent (DSS) as previously performed for the pyroptosome but we have not been able to visualize oligomerization using this technique. Caspase-11 has been shown recently⁶ to be active without oligomerization. Furthermore, LPS in micelle, liposome or in the context of bacterial outer membrane might be the relevant in cellulo ligand for caspase-4 possibly driving oligomerization as reported for AIM2 and the dsDNA scaffold. We thus feel *F. novicida* LPS could activate caspase-4 with or without triggering oligomerization of the purified protein and caspase-4 oligomerization in vitro might not reflect caspase-4 oligomerization happening in cellulo.

To strengthen caspase-4 activation, we now provide a Western blot analysis in which U937 cells ectopically expressing 3xFlag-casp4 demonstrates a clear *F. novicida*-dependent caspase-4 processing (Supplementary Fig. 8a).

The following text has been added in the revised manuscript in addition to a discussion on the oligomerization

"We did not observe clear processing of the endogenous pro-caspase-4 possibly due to the relatively low level of the protein. To circumvent this problem, we over-expressed 3x-Flag-Caspase-4 in U937 cells (Supplementary Fig. 8a). Overexpression of 3x-Flag-Caspase-4 increased *F. novicida*-mediated LDH release (Supplementary Fig. 8b) and allowed a clear detection of caspase-4 processed form in the supernatant (Supplementary Fig. 8a). Importantly, the processed form was detected only upon infection with WT *F. novicida* but not upon infection with the Δ FPI mutant strongly suggesting that caspase-4 processing is associated with cytosolic detection of *F. novicida*"

"Furthermore, oligomerization of caspase-4 in response to under-acylated LPS might be different in vitro and in cellulo possibly explaining the discrepancies between our study and a previous study based on recombinant caspase-4³⁹. Indeed, in cellulo oligomerization of caspase-4 might be triggered by the dense packing of LPS inside bacterial membrane, within liposome or in high molecular mass aggregates as recently described⁵¹. Alternatively, a human specific co-factor might favor in cellulo caspase-4 oligomerization in response to under-acylated LPS, although our results suggest that caspase-4 has an intrinsic ability to trigger inflammasome activation in response to *F. novicida* LPS."

3-In general, tetra-acylated lipid A induced responses in human cells were minimal, which is in contrast to the responses elicited by hexa-acylated lipid A.

We do agree with the reviewer that lipid IVA is less potent in activating hMDMs than hexa-acylated lipidA. Yet, using *F. novicida* infection, our data strongly suggest that sensing of under-acylated LPS in hMDMs is physiological since it is driving the main inflammasome response to *F. novicida* infection. Furthermore, Lipid IVA triggers much stronger responses in human macrophages (up to 1500 pg/ml of IL-1 β) than in murine macrophages highlighting the important species sensitivity differences.

To make this clear, we have added the following sentences in the last paragraphs of the introduction and of the discussion of our revised manuscript:

"While the human response to under-acylated LPS/ Lipid A was not as strong as the response to hexa-acylated LPS/ Lipid A, these results indicate that the human non-canonical inflammasome has a broader sensing ability than its murine counterpart."

"While the human responses to under-acylated LipidA and LPS are lower than the responses to hexa-acylated LPS/ lipidA, they are likely to be important in certain infectious settings as exemplified here with

F. novicida."

4-Knockdown experiments were lacking appropriate positive and negative controls throughout the manuscript (for example Fig 1d, e, and f; Fig 2c, d, and e). Without these controls it is difficult to attribute the observed phenotypes to the RNA silencing.

We now provide the efficacy of NLRP3, caspase-1, caspase-4 knockdown at the protein levels. We failed to find an antibody specific for endogenous human AIM2 in Western blot (as revealed by our CRISPR/cas9 cell line for AIM2 validated both by sequencing and PCR). The non-targeting (negative) siRNA control was (and is) present in each experiment. This might not have been apparent in our former manuscript since the data were normalized to the values of macrophages treated with the non-targeting siRNA.

Regarding the positive controls, caspase-4 siRNA is the best positive control we have for both cell death and IL-1 β and IL-18 release and the corresponding data are presented in Fig. 2. Based on the progression of the manuscript, we cannot present this positive control in Fig.1.

5-Fig 2e. It is surprising that even caspase-1 knockdown did not affect the levels of cleaved caspase-1 in the supernatants.

See reply to reviewer 1 p2: Regarding caspase-1 p20, we now provide in Fig. 2e another western blot with a lower exposure and densitometry-based quantification demonstrating a decrease (58%) in the supernatant. We are very confident that the detected band is specific since it is not detected in U937 cells invalidated for Caspase-1 (see Fig 3h).

We tend to see a greater reduction in pro-caspase-1 level in the cell lysate than of the level of released caspase-1 p20. This is likely due to the fact that a 60 % reduction in procaspase-1 level may not induce a 60% reduction of procaspase-1 recruited to ASC upon inflammasome activation and hence not a 60% reduction in processed caspase-1 released in the supernatant. Of note, we feel the efficacy of caspase-1 knockdown is similar to other studies in the field (see for examples Fig 2D in the study by Sunny shin and collaborators: Casson CN et al. PNAS 2015).

We have added the following sentences in the revised manuscript:

"In agreement with a role of caspase-4 upstream of the canonical inflammasome, knockdown of caspase-4 decreased caspase-1 activation as assessed by immunoblot analysis of caspase-1 p20 subunit in the supernatant (Fig. 3e-88% reduction). As expected knockdown of caspase-4 did not affect pro-caspase-1 level in the cell lysate while knockdown of caspase-1 led to a 62% reduction of pro-caspase-1 protein level. The efficacy of caspase-1 siRNA to decrease the maturation/release of casp1 p20 in the supernatant was slightly lower (58% reduction) possibly due to the number of active ASC complex being more limiting for caspase-1 maturation/release than pro-caspase-1 level. "

6-Did F. novicida LPS transfection/electroporation induce cell death and caspase-1 activation in primary hMDMs?

Fugene-mediated LPS transfection (including *E. coli* LPS transfection) does not lead to robust cell death in primary human macrophages. To provide cell death data upon LPS delivery into the host cytosol we have used electroporation. Cell death upon *F. novicida* LPS electroporation is now provided in Fig. 4D. We now provide in supplemental figure S9b data

demonstrating that this cell death is caspase-4-dependent. Analysis of caspase-1 processing/release by Western blot following *F. novicida* LPS transfection in primary hMDMs is now provided in supplementary Fig. 9a.

The following sentence has been added in the revised manuscript:

"Transfection of *F. tularensis* or *F. novicida* LPS triggered caspase-1 maturation and release (supplementary Fig. 9a) (...)

Fugene-mediated LPS transfection did not elicit robust LDH release in primary hMDMs. In contrast, electroporation of *E. coli* LPS triggered a robust and fast LDH release in both hMDMs and murine BMDMs (in a caspase-11-dependent manner) (Fig. 4D). Importantly, electroporation of *F. novicida* LPS in hMDMs triggered LDH release while it failed to do so in murine BMDMs (Fig. 4D). Knockdown of caspase-4 substantially reduced LDH release upon electroporation of *F. novicida* LPS in hMDMs (supplementary Fig. S9b)."

7-Fig 4a. Did caspase-4 activate caspase-1 and IL-1 β secretion?

We have added the quantification of IL-1 β in Figure 5B (corresponding to the former figure 4) indicating that indeed caspase-4 triggers IL-1 β secretion when ectopically expressed in iBMDMs. The text has been modified to refer to this new piece of data:

"Similarly, expression of caspase-4 rendered iBMDMs able to secrete IL-1 β in response to *F. novicida* LPS electroporation (Fig. 5b)."

8-Fig 5. Did tetra-acylated lipid A transfection induce cell death in hMDMs?

As mentioned above and in the revised manuscript, fugene-mediated transfection does not lead to robust cell death in hMDMs. Due to the uncharged nature of Lipid IV A, electroporation is not efficient enough to observe robust cell death. The following sentences have been added in the manuscript:

"Fugene-mediated LPS transfection did not elicit robust LDH release in primary hMDMs. In contrast, electroporation of *E. coli* LPS triggered a robust and fast LDH release (...) We did not perform electroporation of Lipid A or Lipid IVa since electroporation is not highly efficient to deliver uncharged molecules into the host cytosol."

9-Fig 4a. Cell death induced by chimeric caspase-4 CARD fused to the caspase-11 C-terminal domain was partial compared to full length WT caspase-4.

We agree that the cell death phenotype was only partial upon expression of the chimeric Casp4-CARD-Casp11 protein. At this stage, we cannot explain this partial phenotype. As explained to reviewer 1, this partial phenotype although very consistent (over 8 independent experiments) in terms of cell death in WT iBMDMs was not robust enough by the two other assays (IL-1 β in WT iBMDMs and cell death in *Casp1^{-/-}/Casp11^{-/-}* iBMDMs) to be presented in the revised manuscript. We have thus decided to present only strengthened findings on the intrinsic ability of caspase-4 to sense *F. novicida* LPS when expressed in iBMDMs.

10-Fig S12. It is well documented that IFN-g priming augments E. coli LPS and Simonella induced noncanonical inflammasome activation (Shi et al Nature 2014 and Aachoui, et al Science 2013). Therefore the authors' claim that IFN-g priming specifically boosts inflammasome responses to F. novicida LPS but not E. coli LPS is

surprising and the data presented are not convincing.

We now provide an analysis of the effect of IFN- γ priming at both 4 h and 20 h post-LPS transfection. These data (now presented in supplementary figure 14) are consistent with the literature and demonstrates that IFN- γ accelerates the kinetics of hMDMs in response to *E. coli* LPS while it shows a robust effect on *F. novicida* LPS responses at both 4 h and 20 h post-transfection. We have modified the text accordingly.

"Along the course of our experiments, we noticed that IFN- γ priming enhanced inflammasome activation in response to *F. novicida* infection (Supplementary Fig. 13) or to *Francisella* LPS transfection (Supplementary Fig. 14) in hMDMs while we observed a substantial impact of IFN- γ on IL-1 β release upon *E. coli* LPS transfection only at early time post-transfection."

11-Fig 3c. It is surprising that caspase-1 knockdown only minimally affected IL-1 β secretion in response to F. novicida and E. coli.

Once again, this might be due to the fact that a low number of caspase-1 proteins might be active in the inflammasome complex (pro-caspase-1 protein level might not be the limiting factor). The reduction we observed in IL-1 β is consistent with similar infectious studies performed in hMDMs (see Fig 2E from ⁷ Casson et al. PNAS 2015).

The following sentences have been added to the discussion:

" Our results are based on siRNA-mediated gene expression knockdown in primary hMDMs which typically leads to 40 to 70% reduction in the corresponding transcript level (Supplementary Fig. 3b). These knockdown levels are in agreement with other studies using similar primary human cells^{14,37}. A limitation of this approach is that the total amount of a signaling protein (e.g. procaspase-1) might not be the rate-limiting factor for inflammasome activation, cell death and cytokine production. Yet, all the results obtained in primary hMDMs were validated using CRISPR/Cas9 technology in the human monocytic cell line U937 thereby supporting the siRNA results. "

12-The absolute protein values for cytokine ELISAs (pg/ml or ng/ml) should be presented instead of % of NT control.

The change has been done.

Minor comments

1-Fig 1. The knockdown of NLRP3 and AIM2 should be verified at the protein levels.

The knockdown of NLRP3 is shown at the protein level (Fig. 2C). We have not been able to obtain an antibody specifically detecting endogenous human AIM2. We provide PCR-based evidence of exon deletion for our AIM2 Cas9 endonuclease-mediated mutation (Fig. S4a).

2-Fig. 2e. There was no knockdown of caspase-5 so they shouldn't include it in the data for fig 2c.

We have removed caspase-5-related data from the main manuscript.

3-Fig 4b. Protein expression should be shown.

We do not have an antibody recognizing both caspase-11 and caspase-4 which would allow a direct comparison of the protein levels in the different cell lines. We have thus decided to keep qRT-PCR to directly compare the transcript levels in the different cell lines.

We now provide the Western blot related to Caspase-4 and Caspase-11 expression in Fig. 5d, 5g.

4-Figure citations in the text are not in chronological order.

We have modified the order of the figures to fit the order of the text

Reviewer #3

In this paper Brice et al. propose that caspase 4 is responsible for triggering F. novicida-induced pyroptosis and for activating the NLRP3 inflammasome in human macrophages. Moreover, the authors state that human GBPs synergize with caspase-4 to trigger inflammasome responses to cytosolic under-acylated LPS.

The study of caspase 4/5 is important, as non canonical inflammasome activation is still poorly defined. The putative involvement of GBPs is also interesting. However, in spite the model proposed is appealing, the study displays many weaknesses. In general, the great variability of IL-1beta secretion, coupled with the low number of donors studied in some experiments, makes it difficult to reach convincing conclusions. In addition, in most cases silencing was not efficient. Moreover, many data are not compelling, and the confusion in the supplementary figures (legends missing or messy, figures that do not correspond to the description in the text) doesn't help to follow the message of the study.

We apologize for the mistakes and the confusion. We have answered in details to all the reviewer comments below. Briefly, we have increased the number of patients to strengthen our findings, and demonstrate the statistical significance of our results. We have confirmed all our siRNA data (which efficiency are typical of such human system) with CRISPR/Cas9 invalidation and have added experiments using electroporation of hMDMs to obtain cell death data.

In particular:

1. A crucial point in the ms is the study of the induction of pyroptosis following F. novicida infection. Cell death is measured as PI fluorescence and LDH release. The LDH data however display a very high variability. For instance, in fig 1e, six experiments of AIM2 silencing are shown, three of which display an LDH release of less than 50% compared to the control, whereas in two experiments twice as much LDH compared to the control is released. The same is true for fig. 2d (see LDH release by caspase 5-silenced cells!) and Fig 6c.

The infectious process is a complex one and is particularly sensitive to inter-individual variations. This is particularly striking in an end-point assay such as LDH release. As an example of a similar infectious context in primary human macrophages, you could have a look at the supplemental figures S4 and S7 from ⁷ Casson et al. PNAS 2015. The data presented in our revised manuscript are based on 7 independent healthy donors (Fig. 2c, 3d).

Due to the uncertainty associated with the inter-individual variability, we have validated all the cell death data obtained using siRNA on primary hMDMs with CRISPR/Cas9 invalidated cell lines. Particularly, in the revised version, we present U937 cells invalidated for GasderminD, a key molecule controlling cell death (see Fig. 3g).

LDH release may not be the most robust assay in primary human macrophages but the lack of role of AIM2, NLRP3, caspase-1 and the role of caspase-4 in pyroptosis are backed-up by i) CRISPR/cas9 experiments and ii) analysis of IL-1 β for AIM2 and caspase-4. We are thus highly confident in our data.

We have added Fig. 2f, 2g (LDH assay and real time PI-based cell death assay in AIM2- and

NLRP3-invalidated cell lines), Fig. 3g (real time propidium iodide-based cell death assay in *CASP1*-, *CASP4*-, *GSDMD*-invalidated cell lines) to support our cell death results.

Furthermore, we now present in supplementary Fig. 7, a real time analysis of propidium iodide incorporation/ fluorescence in *F. novicida*-infected hMDMs using two different siRNA for caspase-1, 4 and 5. This experiment although performed on one individual strengthens the role of caspase-4 in promoting *F. novicida*-mediated cell death and demonstrates the robustness of the phenotype.

2.a In general, silencing is highly inefficient (see supplementary figure 2 for NLRP3, AIM2 and MEFV; supplementary fig. 15 for GBPs). Silencing of caspase 1, 4, and 5, shown in figure 2e is also non convincing.

Silencing in primary hMDMs in the literature is typically in the range of what we have observed in our assay (50 to 60 % reduction at the transcript level): see for example Fig 1e in ⁸ Vigano E. et al. Nat. Com 2015 for *NLRP3* silencing or Fig 4 in ⁹ Meixenberger K et al. JI 2010.

While the inflammasome field has been dominated by the study of mouse bone marrow derived macrophages, we feel it is important to move on to study the human inflammasomes. siRNA-mediated knockdown in primary human macrophages will never reach 100% and will demonstrate only partial phenotype especially when targeting enzymes. Indeed the rate-limiting factor for their activity might not be the level of the pro-enzyme but the level of activated enzyme that might not be directly correlated to the former.

Furthermore, all of siRNA have a similar efficacy at the transcript level (we now present all the siRNA on a single figure panel; supplementary Fig. 3b) strongly suggesting that if some demonstrated an effect (e.g. *CASP1*, *CASP4*, *NLRP3* and *GBP2*) while other did not (e.g. *AIM2*), it is very likely to be significant and not artefactual.

We now provide the efficacy of siRNA at the protein level in primary infected hMDMs (77% Fig. 2c for *NLRP3*, 62% Fig. 3e for pro-caspase-1, 60% Fig. 3e for Pro-caspase-4). Unfortunately, we were not able to obtain a specific antibody for the detection of endogenous human *AIM2*.

Finally, all our siRNA results are backed-up by CRISPR/cas9 in a human cell line.

We are thus confident in our conclusions.

We have added the following sentences in the revised manuscript:

"Our results are based on siRNA-mediated gene expression knockdown in primary hMDMs which typically leads to 40 to 70% reduction in the corresponding transcript level (Supplementary Fig. 3b). These knockdown levels are in agreement with other studies using similar primary human cells^{14,37}. A limitation of this approach is that the total amount of a signaling protein (e.g. procaspase-1) might not be the rate-limiting factor for inflammasome activation, cell death and cytokine production. Yet, all the results obtained in primary hMDMs were validated using CRISPR/Cas9 technology in the human monocytic cell line U937 thereby supporting the siRNA results."

2b.Caspase 1 silencing gave surprising results: the intracellular pro-caspase 1 is decreased but the secreted p20 is very high. The presence of secreted p20 explain why IL-1b secretion is only slightly decreased, but does not provide information on the role of caspase 1 compared to the other caspases.

As explained above, we now provide in Fig. 2e another western blot with a lower exposure

demonstrating a decrease (58%) in the supernatant. We further provide densitometry-based quantification of the western blot.

We tend to see a greater reduction in pro-caspase-1 level in the cell lysate than of the level of released caspase-1 p20. This is likely due to the fact that a 60 % reduction in procaspase-1 level may not induce a 60% reduction of procaspase-1 recruited to ASC upon inflammasome activation and hence not a 60% reduction in processed caspase-1 released in the supernatant. Of note, we feel the efficacy of caspase-1 knockdown is similar to other studies in the field (see for examples Fig 2D in the study by Sunny shin and collaborators: Casson CN et al. PNAS 2015).

We have added the following sentences in the revised manuscript:

"In agreement with a role of caspase-4 upstream of the canonical inflammasome, knockdown of caspase-4 decreased caspase-1 activation as assessed by immunoblot analysis of caspase-1 p20 subunit in the supernatant (Fig. 2e-88% reduction). As expected knockdown of caspase-4 did not affect pro-caspase-1 level in the cell lysate while knockdown of caspase-1 led to a 62% reduction at the protein level. The efficacy of caspase-1 siRNA to decrease the maturation/release of casp1 p20 in the supernatant was slightly lower (58% reduction) possibly due to the number of active ASC complex being more limiting for caspase-1 maturation/release than the absolute pro-caspase-1 levels."

We now provide data using *Caspase-4*- and *GasderminD*-invalidated U937 cells, which demonstrate that caspase-1 activation/release is downstream of both Caspase-4 and GasderminD (Fig. 3h). We thus feel the role of caspase-1 in regards to the role of caspase-4 is now clarified. The following sentence has been added in the revised manuscript:

"As previously observed using siRNA-mediated knockdown in hMDMs, this gene ablation technique demonstrated in PMA-differentiated U937 macrophages, a key role of caspase-4 and its downstream target gasdermin D in promoting *F. novicida*-mediated caspase-1 maturation and release (...) (Fig. 3f-h)."

2c. Caspase 5 is not decreased in cas5-silenced cells: again, this data explains the irrelevant effects on cytokine secretion, but does not provide information on the role of caspase 5. Actually, the only silencing that worked was that of caspase 4, that indeed results in decreased cytokine secretion. However, this result does not rule out a role for caspase 5. Moreover, since caspase 1 and 5 have not been efficiently silenced, also the conclusion: "caspase-4 also controlled f. novicida-mediated hMDMs death" (p. 7) is not based on solid data.

We have removed the data related to caspase-5 from the main manuscript since as described previously the knockdown was not efficient at the protein level. We now present data based on ectopic expression indicating that while caspase-5 ectopic expression allows Casp4-invalidated U937 cells to respond to *E. coli* LPS, it does not restore the response to *F. novicida* LPS. (Supplementary Fig. 5)

We have to disagree with the reviewer that our conclusion that caspase-4 controlled *F. novicida*-mediated hMDMs death is not based on solid data.

First, we observed an average 66% reduction in LDH release upon knock-down of caspase-4 compared to a non-targeting siRNA. This experiment is based on macrophages from 7 healthy donors and is highly significant (p-value=0.0006, Fig. 3d).

Second, we have tested two siRNA against caspase-4 demonstrating the same phenotype in terms of cell death. This is now presented in supplementary Fig. 7.

Third, the data is highly consistent with the effect of caspase-4 siRNA on caspase-1 processing and release (Fig. 3e), IL-1 β , IL-1 α and IL-18 release (Fig. 3c).

Fourth, our siRNA data is backed-up by crispr-Cas9 data that are presented in the revised manuscript in Fig. 3f,g, h.

We do not think that the only silencing that worked is the silencing of caspase-4 since we see similar decrease in procaspase-1 levels (62%) upon caspase-1 siRNA and of pro-caspase-4 levels (60%) upon caspase-4 siRNA at the protein levels (see Fig. 2e) and at the transcript level (see supplemental Fig. 3b). We are confident that the major effect of caspase-4 siRNA-mediated knockdown is due to the major and upstream role of caspase-4 in hMDMs in sensing *F. novicida*.

2d. Since specific inhibitors for caspase 1 and 4/5 exist (YVAD and zLEVD), the authors could have used them to substantiate their results that, as such, are not compelling.

Regarding the inhibitors, they are not as specific as described by some vendors. This is well exemplified by the cleavage site of IL-18 (-LESD-), a well-described substrate of caspase-1, which is very close to the so-called specific caspase-4/5 inhibitor (-LEVD-). This is also illustrated by a study by Thornberry et al.¹⁰ presented below:

Fig1: Substrate/Inhibitor specificities of the human inflammatory caspases.

The y axis represents the rate of AMC production expressed as a percentage of the maximum rate observed in each experiment. The x axis shows the positionally defined amino-acids

Following the reviewer's recommendation, we have performed the experiment using inhibitors. We have observed a dose-dependent decrease in IL-1 β levels with both YVAD and LEVD now presented in supplementary Fig. 6 of the revised manuscript. Unfortunately, we did not observe a robust inhibition of cell death possibly due to some toxic effects that we observed at 20 μ M concentrations for both inhibitors. The following sentence has been added in the manuscript:

"The role of the inflammatory caspases in IL-1 β release was strengthened using zYVAD-FMK and zLEVD-FMK inhibitors (Supplementary Fig. 6) although we could not ascertain whether each inhibitor specifically inhibited caspase-1 and/or caspase-4³⁶"

Altogether, we have more confidence in genetic evidence based on siRNA in primary hMDMs and CRISPR/Cas9 in a human macrophage cell line than in the results of tripeptide-based inhibitors.

3. To support the difference between caspase 11 and caspase 4, the authors show that transfection of *F. novicida* and *tularensis* LPS induces IL-1 β secretion in hMDMs (Fig. 3b) but not in BMDMs (Fig. 3a). However, the low level of secreted IL-1 β coupled to the high variability among the 4 donors results in lack of significance. In Fig. 3c, again, the low inhibition of IL-1 β secretion observed in cells silenced for caspase 1 suggests that silencing did not work.

We have increased the number of patients and present now data from 9 mice and 9 healthy controls in Fig. 4a, b in our revised manuscript. Transfection of *Francisella* LPS leads to significant differences ($p=0.0064$, compared to no Fugene) in IL-1 β levels (but not in TNF levels) in hMDMs but not murine BMDMs ($p=0.393$).

Silencing of caspase-1 also leads to a significant decrease in IL-1 β levels in combined independent experiments with 5 healthy donors upon transfection with both *F. novicida* LPS ($p=0.0128$) and *E. coli* LPS ($p=0.0164$) (Fig. 4c of the revised manuscript).

4. Figure 5 shows IL-1 β secretion by human and mouse macrophages transfected with different LPS and lipid A. While it is clear what happens in mice (transfection of LPS *E.c.* and of Lipid A induce secretion), it is very difficult to draw conclusions from the data obtained in human macrophages. These data are too variable, the number of donors too little (3 or 4) and, as such, these results are neither statistically nor biologically significant.

We have now included in Fig. 6a, the values from 7-8 independent healthy donors and have changed our way of presenting the data for a greater visibility. While the data are still variable due to inter-individual variability, the results are now highly significant and strengthen our conclusions.

References:

1. Shi, J. *et al.* Inflammatory caspases are innate immune receptors for intracellular LPS. *Nature* **514**, 187–192 (2014).
2. Kravets, E. *et al.* Guanylate binding proteins directly attack *Toxoplasma gondii* via supramolecular complexes. *eLife* **5**, (2016).
3. Wacker, M. A., Teghanemt, A., Weiss, J. P. & Barker, J. H. High-affinity caspase-4 binding to LPS presented as high molecular mass aggregates or in outer membrane vesicles. *Innate Immun.* **23**, 336–344 (2017).
4. Bekpen, C. *et al.* The interferon-inducible p47 (IRG) GTPases in vertebrates: loss of the cell autonomous resistance mechanism in the human lineage. *Genome Biol.* **6**, R92 (2005).
5. Khaminets, A. *et al.* Coordinated loading of IRG resistance GTPases on to the *Toxoplasma gondii* parasitophorous vacuole. *Cell. Microbiol.* **12**, 939–961 (2010).
6. Zanoni, I. *et al.* An endogenous caspase-11 ligand elicits interleukin-1 release from living dendritic cells. *Science* **352**, 1232–1236 (2016).
7. Casson, C. N. *et al.* Human caspase-4 mediates noncanonical inflammasome activation against gram-negative bacterial pathogens. *Proc. Natl. Acad. Sci. U. S. A.* **112**, 6688–6693 (2015).
8. Vigano, E. *et al.* Human caspase-4 and caspase-5 regulate the one-step non-canonical inflammasome activation in monocytes. *Nat. Commun.* **6**, 8761 (2015).
9. Meixenberger, K. *et al.* *Listeria monocytogenes*-infected human peripheral blood mononuclear cells produce IL-1 β , depending on listeriolysin O and NLRP3. *J. Immunol. Baltim. Md 1950* **184**, 922–930 (2010).

10. Meixenberger, K. *et al.* *Listeria monocytogenes*-infected human peripheral blood mononuclear cells produce. *J. Immunol. Baltim. Md 1950* **184**, 922–930 (2010).
11. Thornberry, N. A. *et al.* A combinatorial approach defines specificities of members of the caspase family and granzyme B. Functional relationships established for key mediators of apoptosis. *J. Biol. Chem.* **272**, 17907–17911 (1997).

Reviewers' comments:

Reviewer #1 (Remarks to the Author):

Brice et al. have thoroughly and thoughtfully addressed the comments in their revised manuscript. This is a novel and interesting story that reveals interesting differences between the way that human and mouse casp4 responds to LPS.

Reviewer #2 (Remarks to the Author):

The revised manuscript has sufficiently addressed this reviewer's comments.

Reviewer #3 (Remarks to the Author):

The authors made several new experiments that actually strengthened many weak data present in the previous version. In my opinion, the new data are not well merged with the figures and the text of the first version. So the revised ms has as much as 18 supplementary Figures are present, is in some points hard to read, and the main message(s) are not clearly focused. The conclusion that caspase 4 and GBPs are involved in inflammasome response to *F. novicida* is probably correct, but the data are not always compelling

Some examples:

I pointed out in my previous comments that data on macrophages transfected with the various siRNA are not convincing as the reduction in mRNA is low. The authors replied that this happens in primary cells. I agree, but then they cannot base their conclusions on this experimental approach, unless they have the evidence that the relevant protein is strongly decreased. In the revised version, they used an alternative approach, CRISP/cas9 in a continuous cell line that of course is different from primary cells but, I agree, is at present the best model they have. However, they maintained the siRNA data. While there is no doubt about the involvement of NLRP3 (NLRP3 protein is actually missing, and caspase-1 activation and IL-1b secretion are inhibited in NLRP3 silenced cells), the exclusion of a role for pyrin (as an example) is not supported by the data. In fact, silencing of MEFV resulted in only 40% decrease of MEFV mRNA; the remaining mRNA could well be translated in a protein that can be functional. No decrease of pyrin is shown, so the conclusion that pyrin is not involved is not supported by the data.

Caspase 4 and 5 represent the human homologues of caspase-11. The specific function of each of them is still unclear. Fig. 3 clearly shows that downmodulation of caspase 4 reduces IL-1b secretion. As for caspase-5, the authors state (p16): "We were unable to assess in a robust manner the functional role of caspase-5 in our experimental setting (...) since the strong induction of caspase 5 expression in hMDM upon infection (supplementary Fig 5a) precluded efficient knockdown of caspase 5 at the protein level using siRNA". In spite of this statement, the author show in supplementary Fig. 5C that silencing of caspase 5 does not affect IL-1b secretion. This is clearly misleading. Suppl. Fig 5e shows that casp4em/em U937cells reconstituted with caspase 5 do not die after LPS F.n. Besides only one experiment has been done, investigating in this experimental setting (rather than in casp5 silenced macrophages), whether or not reconstitution of casp4em/em cells with casp5 rescues IL-1b secretion would have possibly provided indication on the potential different role of caspase1 and 5

We thank reviewers 1 and 2 for their positive opinions on our manuscript. We thank the reviewer 3 for acknowledging that the data included in our revised manuscript has strengthened our conclusions. We have taken into considerations all his additional remarks that we have addressed in our revised manuscript and in details below.

Reviewer #1:

Brice et al. have thoroughly and thoughtfully addressed the comments in their revised manuscript. This is a novel and interesting story that reveals interesting differences between the way that human and mouse casp4 responds to LPS.

Reviewer #2:

The revised manuscript has sufficiently addressed this reviewer's comments.

Reviewer #3:

*The authors made several new experiments that actually strengthened many weak data present in the previous version. In my opinion, the new data are not well merged with the figures and the text of the first version. So the revised ms has as much as 18 supplementary Figures are present, is in some points hard to read, and the main message(s) are not clearly focused. The conclusion that caspase 4 and GBPs are involved in inflammasome response to *F. novicida* is probably correct, but the data are not always compelling*

Compared to our first version, we have made a significant effort to merge the new data requested by the referees in the original figures. For example, the relevant CRISPR/Cas9 validation in U937 and the siRNA experiments in hMDMs have been merged in the figures 2 and 3. The requested cell death data has been merged with the IL-1 β data in figure 4, the requested IL-1 β has been merged with the cell death data in figure 5. The same approach has been made regarding the supplemental information.

Regarding the number of supplementary figures, we believe we comply with Nature policies regarding the availability of data as defined in the Nature Cell Biology editorial "Nothing to hide (data not shown)" Nature Cell Biology, vol.8, number 6, june 2006.

We are glad to see that the reviewer 3 agrees that our conclusions regarding caspase-4 and GBPs are probably correct. These conclusions are based on six different readouts (IL-1 β , IL-18, IL-1 α , cell death, caspase-1 and IL-1 β western blot) with siRNA-mediated knockdown performed on primary cells from up to 8 different healthy individuals and validated with CRISPR/Cas9 in a human cell line.

The data pointed out as not always compelling have been strengthened as described below:

Some examples:

I pointed out in my previous comments that data on macrophages transfected with the various siRNA are not convincing as the reduction in mRNA is low. The authors replied that this happens in primary cells. I agree, but then they cannot base their conclusions on this experimental approach, unless they have the evidence that the relevant protein is strongly decreased. In the revised version, they used an alternative approach, CRISP/cas9 in a continuous cell line that of course is different

from primary cells but, I agree, is at present the best model they have. However, they maintained the siRNA data. While there is no doubt about the involvement of NLRP3 (NLRP3 protein is actually missing, and caspase-1 activation and IL-1 β secretion are inhibited in NLRP3 silenced cells), the exclusion of a role for pyrin (as an example) is not supported by the data. In fact, silencing of MEFV resulted in only 40% decrease of MEFV mRNA; the remaining mRNA could well be translated in a protein that can be functional. No decrease of pyrin is shown, so the conclusion that pyrin is not involved is not supported by the data.

We do believe presenting the data in primary cells is of prime importance.

Besides NLRP3, our results demonstrate the functional efficacy of caspase-1 and caspase-4 siRNA.

As explained below, following reviewer 3's comments, we have decided to remove the siRNA on caspase-5 in primary cells.

Regarding Pyrin, we now provide functional data demonstrating the efficacy of our siRNA. Indeed siRNA-mediated knock down of *MEFV* (the gene encoding pyrin) expression reduces the ability of hMDMs to release IL-1 β in response to TcdB, the best characterized Pyrin inflammasome stimulus. This piece of data is presented in supplemental Fig. S3e. To further strengthen the lack of involvement of the Pyrin inflammasome during *F. novicida* infection, we have generated a U937 cell line knock-out for *MEFV* complemented or not with a *MEFV*-expressing plasmid. The results obtained with this cell line are presented in supplementary Fig. S3f-i and demonstrate that overexpression of *MEFV* increases IL-1 β secretion in response to TcdB but has no effect upon *F. novicida* infection further ruling out that Pyrin is involved.

The following sentences have been added in the result section:

"we did not observe any impact of MEFV knockdown (supplementary Fig. 3b) on F. novicida-mediated hMDMs responses (Supplementary Fig. 3c, d). As expected, MEFV knockdown highly reduced IL-1 β secretion in response to Clostridium difficile toxin B (TcdB) (Supplementary Fig. 3e). The lack of Pyrin implication in sensing F. novicida in human macrophages was confirmed in the human monocyte/macrophage cell line U937. U937 cells over-expressing Pyrin demonstrated increased IL-1 β secretion upon TcdB treatment but similar response as Pyrin-deficient U937 cells upon F. novicida infection or Nigericin treatment (Supplementary Fig. 3f-i)."

We only lack functional validation of siRNA in primary cells for AIM2. As very recently reported by Veit Hornung's laboratory, cytosolic DNA sensing in primary myeloid cells mostly depends on cGAS-STING and not on AIM2 (Gaidt et al., 2017, Cell, November 16, 2017; 171, 1–15) thus impairing a functional validation of the siRNA in primary hMDMs. This reference has been added in the revised manuscript (lines 431-433). Yet, our siRNA data is backed-up by CRISPR/Cas9 in U937 cells, which as reported by Hornung et al. for THP-1 cells, has a functional AIM2 inflammasome. Furthermore, as previously described in the manuscript and the previous rebuttal letter, the lack of AIM2 role in sensing *F. novicida* in human cells has been previously described in two publications. So altogether, the lack of role of AIM2 in sensing *F. novicida* in human cells is demonstrated by siRNA, by CRISPR/Cas9 and is now confirmed by three independent publications.

We thus believe all of our results are robust and support our conclusions regarding a major role of Caspase-4 and NLRP3 in the *F. novicida* inflammasome cascade in human macrophages.

Caspase 4 and 5 represent the human homologues of caspase-11. The specific

function of each of them is still unclear. Fig. 3 clearly shows that downmodulation of caspase 4 reduces IL-1b secretion. As for caspase-5, the authors state (p16): "We were unable to assess in a robust manner the functional role of caspase-5 in our experimental setting (...) since the strong induction of caspase 5 expression in hMDM upon infection (supplementary Fig 5a) precluded efficient knockdown of caspase 5 at the protein level using siRNA". In spite of this statement, the author show in supplementary Fig. 5C that silencing of caspase 5 does not affect IL-1b secretion. This is clearly misleading.

Suppl. Fig 5e shows that casp4em/em U937cells reconstituted with caspase 5 do not die after LPS F.n. Besides only one experiment has been done, investigating in this experimental setting (rather than in casp5 silenced macrophages), whether or not reconstitution of casp4em/em cells with casp5 rescues IL-1b secretion would have possibly provided indication on the potential different role of caspase1 and 5.

To avoid any misleading conclusions, we have removed the statement related to Casp5 siRNA in primary hMDMs and the corresponding supplementary panel.

We have followed reviewer 3's recommendation to assess the role of caspase-5 in Casp4^{em/em} U937 cells ectopically expressing caspase-5. The experiment presented in Fig. S5g (formerly Fig. S5e) has now been repeated three times and we have included IL-1 β data as requested by reviewer 3 (Fig. S5h). Caspase 5 when ectopically expressed in CASP4^{em/em} cells partially rescued the ability of these cells to die and release IL-1 β in response to *E. coli* LPS transfection but not in response to *F. novicida* LPS transfection or to *F. novicida* infection. The following paragraph has thus been included in the revised manuscript:

*"We were unable to assess the functional role of endogenous caspase-5 in our experimental settings since, as previously described⁴⁹, caspase-5 expression was undetectable in the U937 cell line at steady state and since the strong induction of caspase-5 expression in hMDMs upon infection (supplementary Fig. 5a) precluded efficient knockdown of caspase-5 at the protein level using siRNA. Yet, ectopic expression of Caspase-5 partially restored the ability of casp4em/em U937 cells to activate the inflammasome in response to *E. coli* LPS transfection but not in response to *F. novicida* infection or to *F. novicida* LPS transfection (supplementary Fig. 5) suggesting that caspase-5 by itself cannot trigger inflammasome activation in response to cytosolic *F. novicida* LPS."*

REVIEWERS' COMMENTS:

Reviewer #3 (Remarks to the Author):

ok